



# GraphFlood 1.0: an efficient algorithm to approximate 2D hydrodynamics for Landscape Evolution Models

Boris Gailleton[1], Philippe Steer[1], Philippe Davy[1], Wolfgang Schwanghart[2], and Thomas Bernard[1]

[1]CNRS - Geosciences Rennes, Université de Rennes, France
[2]University of Potsdam, Potsdam, Germany

**Correspondence:** Boris Gailleton (boris.gailleton@univ-rennes.fr)

**Abstract.** Computing hydrological fluxes at the Earth's surface is crucial for landscape evolution models, topographic analysis, and geographic information systems. However, existing formalisms, like single or multiple flow algorithms, often rely on ad-hoc rules based on local topographic slope and drainage area, neglecting the physics of water flow. While more physics-oriented solutions offer accuracy (e.g. shallow water equations), their computational costs limit their use in term of spatial and temporal scales. In this conrtibution, we introduce GraphFlood, a novel and efficient iterative method for computing river depth and water discharge in 2D on a digital elevation model (DEM). Leveraging the Directed Acyclic Graph (DAG) structure of surface water flow, GraphFlood iteratively solves the 2D shallow water equations. This algorithm aims to find the correct hydraulic surface by balancing discharge input and output over the topography. At each iteration, we employ fast DAG algorithms to calculate flow accumulation on the hydraulic surface, approximating discharge input. Discharge output is then computed using the Manning flow resistance equation, similar to the River.lab model (Davy and Lague, 2009). Iteratively, the divergence of discharges increments flow depth until reaching a stationary state. This algorithm can also solve for flood wave propagation by approximating the input discharge function of the immediate upstream neighbours. We validate water depths obtained with the stationary solution against analytical solutions for rectangular channels and the River.lab and Caesar Lisflood models for natural DEMs. GraphFlood demonstrates significant computational advantages over previous hydrodynamic models, with approximately a 10-fold speed-up compared to the River.lab model (Davy and Lague, 2009). Additionally, its computational time scales slightly more than linearly with the number of cells, making it suitable for large DEMs exceeding $10^6$ - $10^8$ cells. We demonstrate the versatility of GraphFlood in integrating realistic hydrology into various topographic and morphometric analyses, including channel width measurement, inundation pattern delineation, floodplain delineation, and the classification of hillslope, colluvial, and fluvial domains. Furthermore, we discuss its integration potential in landscape evolution models, highlighting its simplicity of implementation and computational efficiency.

## 1 Introduction

River dynamics encompass key processes of landscape evolution at different temporal and spatial scales. Rivers transfer sediments downstream, they control the baselevel of hillslopes, and set the pace of denudation rates (e.g. Clubb et al., 2019). Modelling landscape evolution and the development of fluvial landforms, in particular, thus requires a sound representation of





how rivers erode, transport and deposit material. As landscape evolution models are used to simulate the dynamics of topogra-
phy over $10^5$-$10^7$ years and at continental scales (Salles et al., 2023), accounting for short-term processes (e.g. daily variations
of discharge, flood) at local scales remains a methodological and numerical challenge. Simulating flow in open environments
in two or three dimensions requires sophisticated numerical methods which are computationally demanding and which are
thus mostly inapt for the challenge of simulating landscape evolution over geological time scales (Davy et al., 2017). Instead, a

common approach to model water flow across landscapes consists in applying the single or multiple flow algorithms (e.g. Tar-
boton, 1997; O'Callaghan and Mark, 1984). These techniques route water along topographic gradients towards one or multiple
neighboring pixels in a DEM and approximate discharge by drainage area weighted by precipitation rates (Adams et al., 2020).
The approximation of steady flow using drainage-area based discharge has been the cornerstone of integrating hydrodynamics
in long-term erosion laws (e.g. Whipple and Tucker, 1999). This approach has the compelling advantage that it reduces flow

patterns to a network of flow lines, and has been widely used to establish empirical scaling laws relating drainage area to
channel steepness and uplift (Wobus et al., 2006), or to unravel landscape evolution from the planform shape of the river net-
works (Schumm et al., 2000; Willett et al., 2014). Moreover, these methods rely on efficient algorithms, which leverage graph
theory to compute drainage area (e.g. Braun and Willett, 2013; Anand et al., 2020), flow across complex terrain (e.g. Barnes
et al., 2014; Cordonnier et al., 2018; Barnes et al., 2021; Schwanghart and Scherler, 2017) or geomorphological metrics (e.g.

Gailleton et al., 2019; Mudd et al., 2018; Grieve et al., 2018; Schwanghart et al., 2021). In particular the Single Flow Direction
(SFD) algorithm is thus the numerical workhorse for simulation software for landscape evolution (Hergarten, 2020; Braun and
Sambridge, 1997; Willgoose et al., 1994; Campforts et al., 2017; Braun and Willett, 2013, e.g.) and numerical frameworks for
quantitative geomorphology (e.g. Barnhart et al., 2020; Gailleton et al., 2023; Schwanghart and Scherler, 2014; Mudd et al.,
2019).

45       However, reducing rivers to lines in landscape evolution models may overtly simplify the dynamics and feedbacks of fluvial
processes (Armitage, 2019). In fact, the response of rivers to climate variability, tectonic movements or baselevel changes is
more varied than the simple propagation of a wave of vertical changes through 1D network of lines. For example, changes
in boundary conditions cause rivers to adjust their width (e.g. Dunne and Jerolmack, 2020; Baynes et al., 2022) and their
planform flow pattern (e.g. Schuurman et al., 2013), both of which feedback on sediment fluxes (e.g. Davy and Lague, 2009).

In addition, the past decade has seen the rising availability of high resolution lidar-derived DEMs (<1 m resolution). This means,
however, that for a variety of geomorphological applications (e.g. Steer et al., 2022; Stammberger et al., 2024) rivers cannot be
realistically represented by one pixel-wide paths (Figure 2). Several recent studies demonstrate the advantages of integrating
2D hydrodynamics to inform the study of landforms (Costabile et al., 2019; Costabile and Costanzo, 2021; Bernard et al.,
2022), even on long timescales. Here, we present a new and efficient method, based on graph theory and finite differences, to

fill this methodological gap and allow the efficient approximation of 2D hydrodynamics on high resolution topography and/or
longer term landscape evolution model.



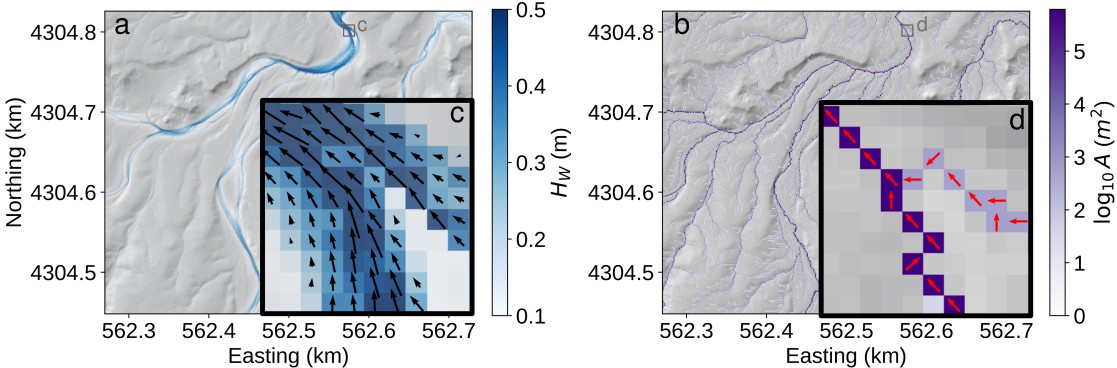

**Figure 1.** Comparison between water flows approximated with GraphFlood (a and c), calculating flow depth and discharge vectors, and with a classic drainage area based method (D8 Steepest descent route) (b and d). The panels detail a channel junction and highlight how GraphFlood models flow patterns and how these differ from one-pixel wide flows derived from the D8 algorithm.

## 1.1 Existing solutions

A range of numerical models incorporating 2D to 3D hydrodynamics to study river systems and their morphological evolu-
tion exists, with widely different methods and levels of complexity, depending on the temporal and spatial scales of interest.
Finite-element models are commonly used for reach-scale models, such as DELFT3D (Roelvink and Banning, 1995), HEC
RAS (Brunner, 2002), BASEMENT (Vanzo et al., 2021) or TELEMAC (Villaret et al., 2013). These models are designed for
simulating the evolution of fluvial landforms over scales of 1-100 km and over 1-100 years, and therefore fall outside the scope
of this study.

Bates et al. (2010) developed a two-dimensional hydrodynamic model Lisflood-FP, solving for the 2D Shallow Water Equa-
tions (SWE). Their cellular-automata approach has been successfully incorporated in the landscape evolution model CAESAR
Coulthard et al. (2013) to simulate reach-to-catchment scale fluvial hydro-morphodynamics (e.g. Yu and Coulthard, 2015;
Liu and Coulthard, 2015; Coulthard and Van De Wiel, 2017). Lisflood-FP adopts a finite difference scheme on the bidirec-
tional water fluxes between pixel. While it has been applied to catchment scales over potentially thousands of years (Liu and
Coulthard, 2017, e.g.), its potential for longer-term and larger-scale studies remains hampered by the physics behind which
explicitly simulates wave propagation. Indeed, any upstream change of runoff input (e.g., precipitation) needs to be gradually
propagated downstream one pixel per computational time step. While modelling non-steady flows is important for simulating
transient responses to individual storm events (e.g. Van De Wiel and Coulthard, 2010), it represents a limiting factor aiming for
simulating longer time scales. Bates et al. (2010) and subsequent improvements by de Almeida et al. (2012) have been utilized
in other landscape evolution framework (e.g. Barnhart et al., 2020) following the same principle.

An alternative to propagating wave is to focus on the stationary state of the river network (i.e., in equilibrium with the input
field of runoff). The main challenge in estimating efficiently the stationary solution lies in spreading the flow to its equilibrium



field. The latter depends on the final geometry of the hydraulic surface, which cannot be deduced from the geometry of the terrain alone. To address this point, Davy et al. (2017) developed an efficient particle-based solution to solve the SWE. In this approach, precipitons (i.e., elementary volumes of water) are dropped on the landscapes and propagate following a stochastic

path down the hydraulic surface. Precipitons increase the water height along their path, bypassing the need to to propagate flood waves gradually. The frequency at which precipitons pass a cell determines the amount of water received by this cell, balanced by a decrease of flow depth based on discharge calculated with Manning's equations. This method is efficient in terms of computation time (Davy et al., 2017), and in particular in the fluvial domain having high frequency of precipiton passage. However, it has some physical and numerical drawbacks: i) each precipiton is on a different timeline making the isolation of

snapshots through time challenging; ii) the fluvial domain receives many more precipitons than the hillslope domain, making their repeated passage numerically redundant while displaying slower convergence time on hillslopes; and iii) precipitons are independent one from another and only integrate information down their 1D flow path. A similar approach has been developed by Pelletier (2008), who outlined the prototype of a highly-iterative solution that repeatedly runs the MFD model on the terrain and the water surface. This process incrementally increases the flow height until satisfying an equilibrium between flow depth

and input discharge. This approach is the starting point for our new algorithm.

## 1.2    A new solution based on graph theory

GraphFlood uses a novel approach to efficiently calculate the stationary solution for the whole landscape. Topography can numerically be described as a data structure where each location of a DEM is linked to its neighbours *via* unique directional connections *upstream* or *downstream*. In graph theory, this data structure is called a Directed Acyclic Graph (DAG) and opens a

range of efficient algorithms applied to the propagation of information through a landscapes (see the review work of Heckmann et al., 2015). We leverage the DAG nature of the topography to propagate runoff through the whole landscape at every single time step using drainage area calculated on the hydraulic surface. Using the DAG structure, calculating drainage-area is very efficient and can be done in a single graph traversal following the downstream topological order (e.g. Anand et al., 2020; Braun and Willett, 2013; Gailleton et al., 2023; Hergarten and Neugebauer, 2001). Weighted by precipitation rates, drainage area

determines the amount of water entering every cell of the system. At each iteration, we calculate the discharge leaving the cells following a SWE, neglecting inertia (Davy et al., 2017). The balance of the input and output discharges iteratively increments flow depth until reaching an equilibrium of the water surface.

In the following, we first describe the theory behind our method, before explaining the algorithm and the associated finite difference scheme. Different case studies are then tested to demonstrate the potential of the method for flood modelling,

morphometric analysis, and landscape evolution modelling. Last, we discuss the limitation and next developments for the model.





## 2  Theoretical background

### 2.1  Shallow Water Equations

We use the 2D SWE to approximate the physics of water flow in open-environment. The equations are derived by integrating the three-dimensional Navier-Stokes equations over the vertical dimension, assuming that the velocity field varies primarily in the horizontal direction, and are commonly used to model flooding beyond reach scale (Bates, 2022). The 2D SWEs consist in a mass conservation equation and a momentum conservation equation. Using the notations of Davy et al. (2017), the mass conservation equation can be written:

$$\frac{\partial h}{\partial t} - \nabla \cdot (\boldsymbol{q}) = 0 \tag{1}$$

$h$ is the water depth in [L], $t$ the time in [T] and $q$ the discharge per unit width in $[\frac{L^2}{T}]$.

Neglecting inertia, (Manning et al., 1890) demonstrated that the momentum equation can be simplified into Manning's equations where flow velocity $u$ (in $\frac{L}{T}$) is expressed as:

$$\boldsymbol{u} = \frac{h^\alpha}{n} \frac{\boldsymbol{s}}{\|\boldsymbol{s}\|^{\frac{1}{2}}} \tag{2}$$

where $\alpha$ is Manning's exponent, usually assumed equal to $\frac{2}{3}$, $n$ is Manning's friction coefficient, $xmax$ being the direction of the steepest hydraulic gradient.

In order to insert equation 2 into equation 1, discharge per unit width and velocity are related *via* flow depth:

$$\boldsymbol{q} = \boldsymbol{u} \cdot h \tag{3}$$

Unlike similar methods (Bates et al., 2010, e.g.) or more sophisticated formulations (e.g. Brunner, 2002) incorporating additional physical elements (e.g. inertia, turbulence), our method is designed to be optimized when these components can be neglected (Davy et al., 2017). We use $\boldsymbol{Q}$ to refer to the volumetric flux in $[\frac{L^3}{T}]$ and the indices $\boldsymbol{X_{in}}$ and $\boldsymbol{X_{out}}$ to refer respectively to quantities *entering* or *leaving* a given cell.

These equations can simulate the propagation of water through space and time dynamically, solving a transient flood wave. $\nabla \cdot \boldsymbol{q}$ is the difference between $q_{in}$ made of $q_{out}$ from upstream neighbours and $q_{out}$ from the current cell to its downstream neighbours. For a constant input of $q_{in}$ on a landscape (e.g. constant precipitation rates, fixed input discharge), the system has an equilibrium state - or stationary solution - where the water depth and hydraulic slope lead to a $q_{out}$ balancing $q_{in}$. The total $Q_{in}$ for the stationary state for a given location becomes the integration of all the source terms (e.g. precipitations, resurgence) over the drainage area upstream of a given location.

In this contribution, we refer to the *transient solution* when we seek to solve the transient propagation of $Q$ through space and time and to the *stationary solution* when we are only interested in the equilibrated fields.



## 3   A graph-based iterative method

As stated in section 1.1, there are multiple ways to numerically solve for the SWE. Our developed scheme applies an explicit finite difference scheme on a graph (Braun and Willett, 2013; Barnhart et al., 2020; Gailleton et al., 2023). It aims to provide a reasonably efficient and scalable solution suitable for large-scale DEMs and LEMs. Our iterative scheme is optimised for the stationary solution, but can be used for transient simulation. In the following, we detail the numerical graph structure (DAG) required by our method, we describe the finite difference scheme, explain the transient and stationary solutions and validate them against analytical solutions.

### 3.1   Numerical structure

We use the following terms adopted from graph theory (see  Heckmann et al., 2015, for a comprehensive review about the use of graph theory applied to geomorphological applications): a discrete location is represented by a *node*, linked to its *neighbor* nodes via *links*. The links are directed edges linking *donors* to their downstream *receivers*. In our referential donors have higher hydraulic surface $(Z + h)$ than their receivers. The algorithm is compatible with any type of grid (e.g. hexagonal grid or triangular network), as long as the DAG structure defines the topology between the pixels or facets. Each link is characterized by a specific length $\partial l$ representing the distance between the two neighbour nodes and a link width $\partial w$ representing the local width. Each node represents a cell area $A_c$. The scheme also requires common DAG algorithms: the *topological ordering* - an operation providing a list of nodes sorted from upstream to downstream and *sink filling* a method filling local minimas disconnected from the rest of the graph (e.g. lake, local noise). The DAG can use both Single Flow Direction (SFD) topology (Braun and Willett, 2013), where each node has a single receiver (e.g. steepest descent or D8), or Multiple Flow Direction (MFD) DAGs (e.g. Tarboton, 1997; Anand et al., 2020). This distinction is important as most common operations on SFD DAGs are simpler and more efficient than the MFD DAGs (e.g. Braun and Willett, 2013; Anand et al., 2020). It is worth noting the latter catches more details about flow topology and tend to increase the accuracy of the represented processes (Armitage, 2019, e.g.).

In this contribution, we developed the method for regular grids. In the stationary case, we use the algorithms of Barnes et al. (2014) and Cordonnier et al. (2018) to ensure flow continuity and proceed to an initial filling of the local minimas (e.g. noise, lake). Topological sorting operations use a modified version of Braun and Willett (2013) for SFD and a variant of Anand et al. (2020) for MFD. The modifications are minor changes of data structure that do not change the overall functioning while improving performance and readability (see Gailleton et al. (2023) for detailed implementations). One advantage of GraphFlood is that it can be implemented using existing computational frameworks for DEM analysis and LEM simulation (e.g. Schwanghart and Scherler, 2014; Gailleton and Mudd, 2021; Barnhart et al., 2020). A notable difference compared to existing framework is that we calculate the DAG using the hydraulic surface rather than the topography.

 

## 3.2 Iterative explicit finite difference scheme

We use an explicit finite difference scheme to solve equation 1. In the transient case, the numerical solution predicts flow depth change for every node $i$:

$$\frac{h_i^{t+1} - h_i^t}{\Delta t} = \frac{\sum\limits_{d=donors(i)} Q_{in_d} - \sum\limits_{r=receivers(i)} Q_{out_r}}{A_c} \tag{4}$$

where $Q_{in_d}$ represent the discharge from a donor $d$ to the node $i$ and $Q_{out_r}$ the discharge from the node $i$ and a receiver $r$. For the latter, in the case of SFD (i.e. single receiver), equation 3 becomes:

$$Q_{out_i} = \frac{\Delta W}{n} h_i^\alpha \sqrt{s_{ir}} \tag{5}$$

where $i$ and $r$ are respectively a given node and its single receiver and $\Delta W$ the flow width in the given direction. Because flow can only go through one link, $\Delta W$ is easy to determine. For example for our case of a regular grid, it is $\Delta x$ in the $y$ direction, $\Delta y$ in the $x$ direction and the diagonal length for the other cases. As noted by Coulthard et al. (2013), MFD can become increasingly more complicated: multiple receivers mean $\Delta W$ "overlaps" and using the direct width of flow for each links can break the conservation of mass. Let's imagine a regular grid considering D8 neighbouring (cardinal and diagonal directions), a node that would discharge to all these directions would integrate twice the total flow width. Porting this formulation to MFD requires then a correction factor. Equation 3 in MFD DAG therefore becomes:

$$Q_{out_i} = \frac{C}{n} h_i^\alpha \frac{\sum\limits_{j\,in\,receivers} s_{ij} \Delta W_{ij}}{\sqrt{S_{ijmax}}} \tag{6}$$

By definition, for a given flow depth, both SFD and MFD discharge should be equal. Therefore, the correction factor is:

$$C = \frac{s_{ijmax} \Delta W_{ijmax}}{\sum\limits_{j\,in\,receivers} S_{ij} \Delta W_{ij}} \tag{7}$$

The magnitude of $Q_{out}$ flux is the same for MFD and SFD schemes, but the correction factor states the discharge need to be parted to multiple receivers proportional to $S_{ij} \Delta W_{ij}$.

Both transient and stationary solutions follow that scheme to calculate the output discharge, the difference is the calculation $Q_{in}$ for all nodes. The overall process is outlined on algorithm 1.

## 3.3 Transient solution

For the transient solution, $Q_{in_{di}}$ is $Q_{out_{di}}$ calculated between the donor and this node plus an eventual local external $Q_{in}$ source term (*e.g.,* resurgence, precipitation, grid edge input). The method becomes similar to (Bates et al., 2010) - only that





---

**Algorithm 1** Iterative stationary solver

---

    Initialise DAG structure on hydraulic surface

    **while** Convergence criterion[1] not met **do**

        Update DAG with hydraulic surface

        **for** each node n in downstream topological order **do**

            Calculate $s(n)$ and weight partitioning

            Determine $Q_{in}(n)$ from upstream nodes

            Calculate $Q_{out}(n)$

            Transfer $Q$ to receivers of $n$

        **end for**

        Increment $h_w$ for all nodes

    **end while**

---

their formulation includes an approximation of inertia and have a D4 flow topology. Although straightforward and massively parallelisable (e.g. Apel et al., 2022), this method does not benefit from the DAG data structure as signals are propagated from one node to their immediate neighbours. If external $Q_{in}$ is kept constant long enough, this solution converges toward a unique equilibrium stationary state and is not efficient if the intermediate transient steps are not important.

Like any explicit finite difference methods, higher time steps leads to less iterations and more efficient spread, but also more instability. Equation 6 expresses the velocity of a flood wave and therefore its stability can be approximated using the Courant Friedrich Levy conditions (CFL):

$$C_r = \Delta t \frac{u_{max}}{\Delta x max} \tag{8}$$

where $C_r$ is the Courant number.

The transient solution converges toward an equilibrium hydraulic surface and $Q$ field. We estimate convergence based on both median $h$ and $\frac{\Delta h}{\Delta t}$ for the whole landscape. We stopped the iterative process once the first plateaus and, when increment in flow depth becomes lowerthan an acceptable *ad hoc* threshold (*e.g.* $10^{-9}$ m).

### 3.4 Stationary solution

The stationary solution optimises convergence towards the equilibrated solution - *i.e.* the steady state flow depth and discharge fields to an input runoff. Ultimately, the amount of water flowing through a landscape equates the runoff rate propagated into the drainage network. Numerically speaking, it falls down to calculating a weighted drainage area, a procedure already in use in GIS applications and LEMs when it comes to integrating the effect of spatial variations in precipitations (Leonard et al., 2023, e.g.). In the case of effective precipitations, each nodes receive a local $P(x,y)\Delta x \Delta y$, while in reach mode, given entry nodes receive an arbitrary $Q_{in}$. In both cases, received water is then recursively transferred to all the downstream nodes following the topological order. It effectively reduces the need to propagate a signal gradually from upstream to downstream one node





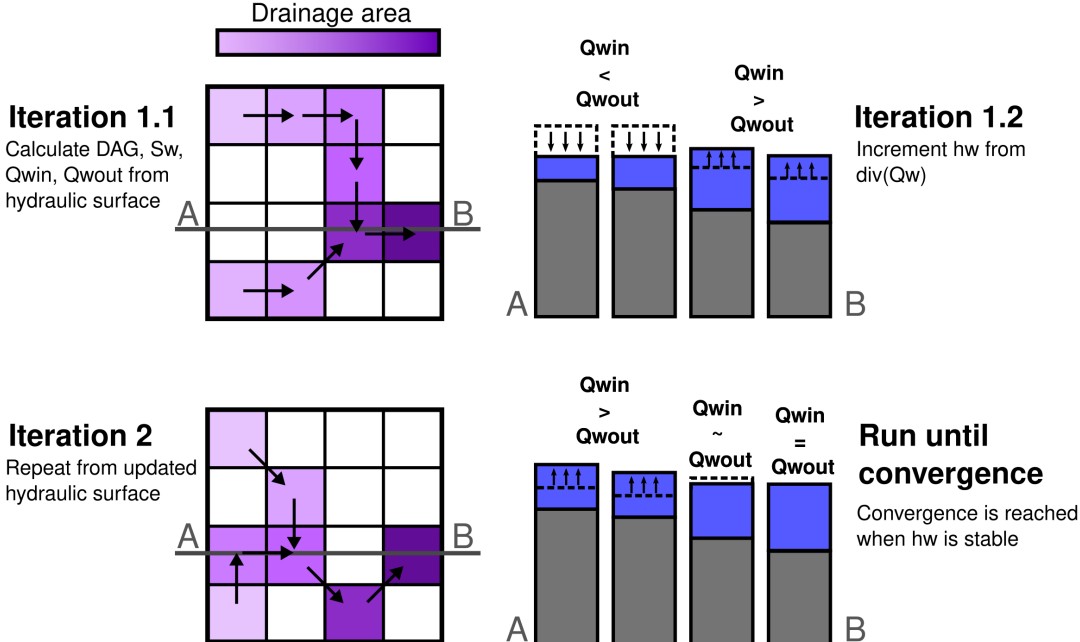

**Figure 2.** Comparison between hydrology approximated with GraphFlood (a and c), calculating flow depth and discharge vectors, and with a classic drainage area based method (D8 Steepest descent route). The pannels zoomed on a channel junction highlight how GraphFlood allows the extraction of detailed flow pattern in all direction and magnitude compared to the D8, linear networks of drainage area.

at a time. However, the final hydraulic surface being different than the topographic surface, the algorithm needs to iterate to gradually build the hydraulic surface. From the first iteration, discharge is propagated through the full landscape and starts "piling up" $h$ on the whole flow path. Every iteration recomputes the DAGs from the updated hydraulic surface, effectively spreading $Q_{in}$ towards its final geometry balanced by $Q_{out}$. Time step in the stationary mode is a numerical stability criterion modulating the magnitude of flow depth increment. Because $Q_{in}$ is independently determined from $Q_{out}$, the CFL stability criterion does not strictly apply and we test the model with a constant or a variable time step (then determined in respect to CFL conditions). Similarly to the transient solution, we estimate convergence based on both median $h$ and $\Delta h$ between each iterations for the whole landscape and considered convergence reached once median $\Delta h < 1e - 9$ m.

### 3.5 Validation

We validate the numerical scheme against an analytical solution (Figure 3) in the case of a rectangular channel (Bates et al., 2010; Davy et al., 2017). We combine equation 1 and equation 3 to obtain an analytical stationary flow depth noted $h_W^*$:





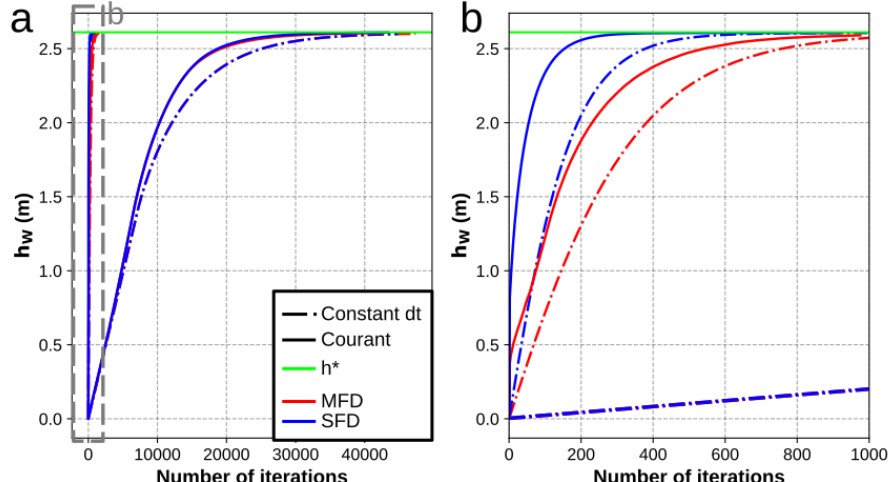

**Figure 3.** Validation tests for the MFD and SFD stationary and transient simulation for a given $Q_{in} = 15 \, m^3.s^{-1}$. The scenarios with constant $dt$ were set to $1e^{-3}$ seconds and the scenarios with CFL condition were calculated with $C_r = 3e^{-3}$. Both were chosen empirically as values balancing model performances, stability and cleanness of the final results. Panel a displays the full results for all the simulations while b zooms on the stationary model results.

$$h^* = \frac{nQ_{in}}{dx\sqrt{s}}^{\frac{1}{\alpha}} \tag{9}$$

Equation 2 predicts that in the case of a rectangular channel with a constant slope $S_0$, the slope of the water surface $s$ should be equal to $S_0$. Assuming a boundary condition of fixed hydraulic slope equals to $S_0$, we can determine $h^*$ suitable for an analytical calibration.

We run GraphFlood with the transient and stationary solvers, and MFD and SFD schemes on a $200 \, m \times 40 \, m$ rectangular channel with a regular $dx = 1 \, m$ (more details in the figure caption). Figure 3a shows the results for all runs. Each simulation converges towards $h^*$, validating the numerical methods. The number of iterations to reach $h^*$ - directly linked to the computational efficiency of the algorithms - is significantly higher for the transient model as it needs to propagate the flood wave through the whole channel one node per iteration. This behaviour is likely to worsen with the complexity of a natural river network where any junction would need catchment-wise upstream information before being equilibrated and being able to propagate signal downstream. Figure 3b zooms on the stationary models that reach stationary state in about 300-1000 iterations, roughly 400 times faster than the transient model. Adaptive time stepping based on the CFL condition slightly reduces the number of iterations required to reach the analytical solution and the SFD model converges in less iterations than the MFD model.



## 3.6 Test sites

We test GraphFlood on two lidar-derived DEMs and aim to explore the effect of different geographical contexts on the algorithm, both in term of relief and climate. Our first test site is located near Green River (Utah, USA), a low-relief area in an arid context with smooth hillslopes. The second test site is the Hanalei river catchment in Hawai (USA), with sharp relief made of volcanic rocks, steep hillslopes and entrenched valleys. The original spatial resolution of both DEMs is 1 m, provided pre-processed from point clouds and provided by opentopography.org (OpenTopography, 2020, 2012). We also downsample the

DEM of the Hanalei river catchment to a resolution of 5 m using a cubic resampling implemented by GDAL/OGR contributors (2023) to process a larger watershed and test GraphFlood on multiple resolutions.

## 4 Results

### 4.1 Numerical behavior for a single simulation

We first explore the behavior of the model during a single simulation, where we run the MFD stationary algorithm on both

test sites for a high-intensity rainfall rate of $100\,mm\,h^{-1}$. We deliberately chose an extreme rainfall rate to test the algorithm under high flow conditions during which multiple diverging river channels are activated.

We run the model to convergence (figure 4 - see caption for the full simulation parameters). In term of channel network topology, GraphFlood is able to reproduce diverging and converging flow patterns that follow converging and diverging channel networks. This behaviour is striking on Green River, where the broad valleys consist of an interwoven network of channels,

but also well-captured on the clearer channel beds of Hanalei. GraphFlood in that way contrasts with drainage-area based flow patterns which by nature converge toward a single line of flow (e.g. fig. 2). In both cases the majority of the DEM pixels are displaying insignificant flow depth (<1 cm) as one should expecting from natural landscapes where rivers only represent small portions of the landscape.

GraphFlood reaches convergence in respectively 4000 and 3000 numerical iterations for Green river and Hanalei (fig. 5

a and b) based on the criterion outlined in sections 3.3 and 3.4. At first glance, this number is high, but we observe a huge discrepancy in the spatial and temporal patterns of convergence. The model converges asymptotically in the rivers where less than 200 iteration for Green River and less than 60 for Hanalei are enough, as illustrated by the striking spatial variations on figure 5 c and d. Low drainage area on the hillslopes induces lower increments of flow depth, which combined with high slopes explain the slower convergence on the hillslopes.

We test the sensitivity of the model to its numerical parameter $\Delta t$ and its discretisation $\Delta x$. $\Delta t$ controls the magnitude of $h$ increment. Maximising it optimises the spreading of $Q$ to its equilibrium field. However, our tests also highlight that while significant over-estimation provokes numerical divergence, slight overestimation converges to an underestimated final $h$. Spatial resolution of DEM, $\Delta x$, can be dictated by the availability of source data, but it can be interesting to reduce the resolution of a DEM in order to process larger area (if computing speed or memory are limiting factors). For this test, we use

the Green River DEM resampled from $dx = 1\,m$ to $dx = 10\,m$. Flow patterns remain relatively similar from a resolution



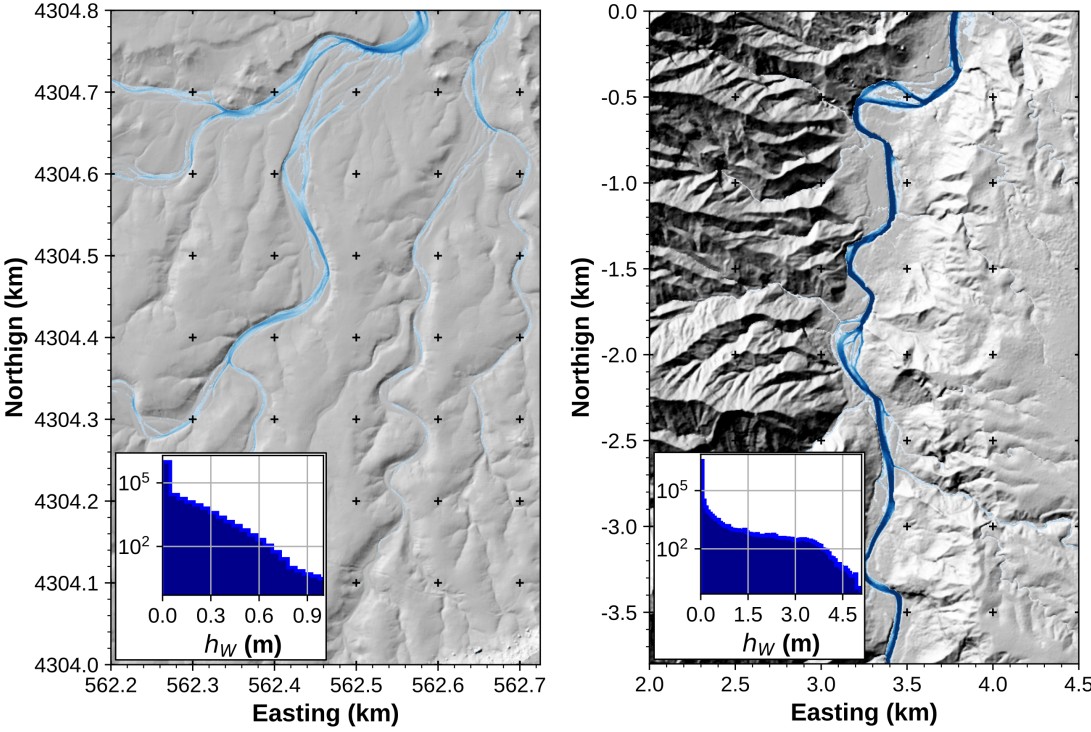

**Figure 4.** Flow depth field calculated with GraphFlood for fluvial valleys in Green River, Wyoming, USA (a) and Hanalei, Hawaii, USA (b). The maps are zoomed on major fluvial valleys for clarity. Both histograms show the distribution of water height for the MFD stationary solutions calculated during a high storm event (precipitation rate = 100 mm/h). Note the logarithmic y scale on the histogram demonstrating the huge majority of points have low flow depth (< 1cm).

to another. However loss of details are observed at lower resolution as expected. Lowering resolution leads to lower hydraulic slopes on averaged and subsequently a decrease of $Q_{out}$ and an increase of total volume of water stored on the DEM.

We also test the sensitivity to the physical parameters. Manning's coefficient is an empirical friction parameter reflecting the local surface condition (e.g. vegetation, bed roughness, see Arcement and Schneider (1989) for different measurements).

Higher friction values predicts a higher and more distributed water surface required to reach the same $Q_{out}$. Higher input discharge or precipitation rates lead to higher flow velocity and therefore lower the stability condition, thus impacting speed of convergence.





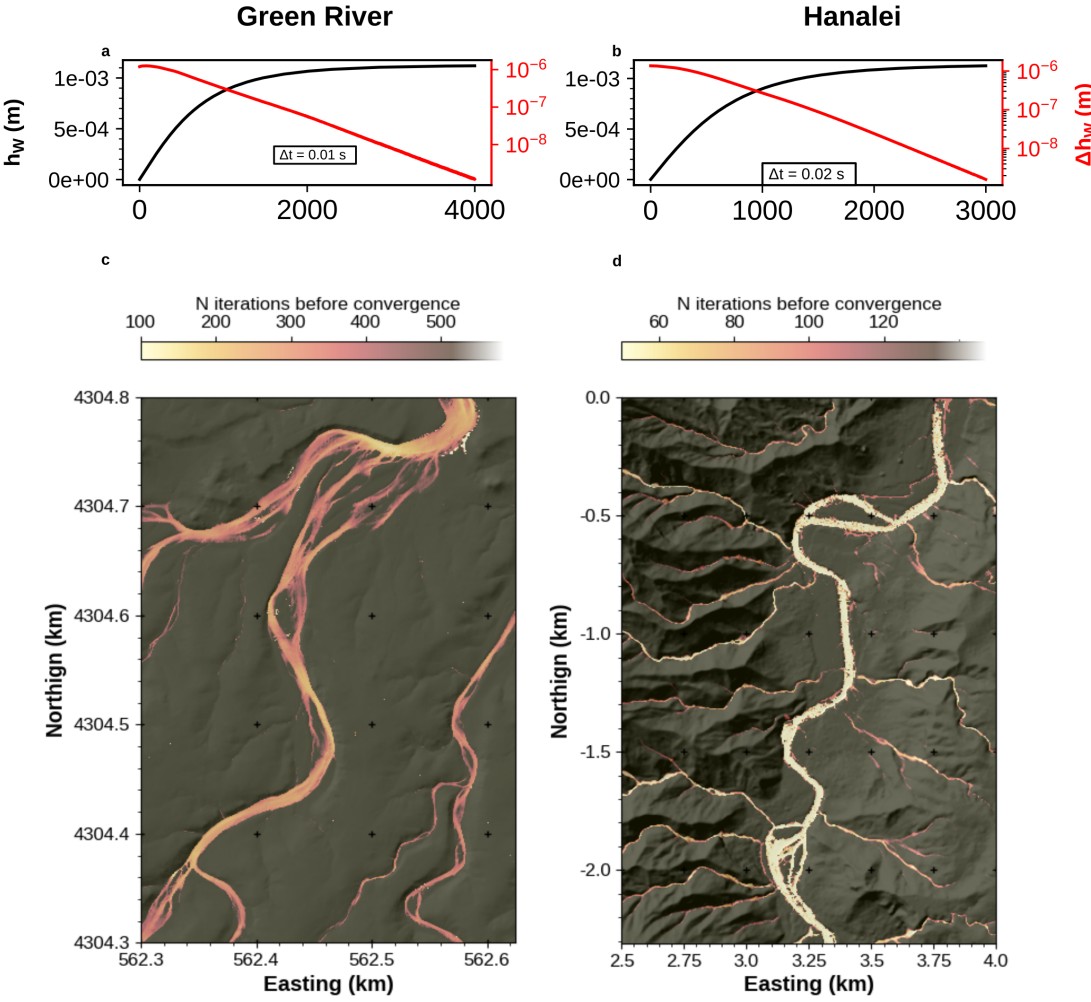

**Figure 5.** Rate of convergence for the simulation of figure 4 with respectively $\Delta t = 1 \times 10^{-2}\ s$ and $\Delta t = 2 \times 10^{-2}\ s$. On panels a and b, we show in black the median flow depth function of the number of numerical iterations and in red the changes in flow depth between each iterations. Panels c and d demonstrate the spatial variability in the rate of convergence. Note that GraphFlood converges significantly faster in fluvial domain. The number of iterations before convergence is defined as the first iteration reaching 95% of its equillibrium value .

## 4.2 Comparison with existing models

We compared GraphFlood with previoous models sharing similar applications (relatively large-scale and medium term hydrol-
ogy): Caesar Lisflood (Coulthard et al., 2013) and River.Lab (formerly Eros/Floodos - Davy et al. (2017)). We ran the three
models on Green River with a constant rainfall rate of $30\ mm\ h^{-1}$ and a classical friction coefficient of $0.033$. We ran the three



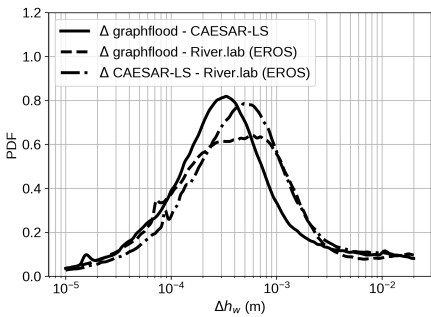

**Figure 6.** Benchmark comparing the difference in stationary field of flow depth between CAESAR-LISFLOOD, River.Lab (formerly EROS/FLOODOS) and GraphFlood. The data expresses the distribution of flow depth differences for each pairs of the models. The distributions are estimated using a Kernel Density Estimation.

stationary simulations, as detailed in section 3.4. We compared the fields of flow depth by pairs of models (figure 6). Overall, the differences between the models are minimal, centered between $3 \ 10^{-4}$ and $5 \ 10^{-4}$ m. The differences can be linked to the differences in flow routing. Caesar Lisflood can only route flow to cardinal directions therefore the distribution of slopes is not exactly the same than GraphFlood and River.lab which include diagonals. River.Lab relies on a stack of consecutive 1D stochastic paths on a 2D grid while GraphFlood offers a continuous solution in space and time, explaining the small differences in the final solutions.

## 5 Applications and potential

### 5.1 Flood extent

The computational efficiency of GraphFloods enables the rapid simulation of stationary flow depth and extents under different runoff intensities. We ran the model for effective precipitation rates ranging from $5 \ mm \ h^{-1}$ - approximating low-flow conditions - to $300 \ mm \ h^{-1}$ - extreme storm conditions. Figure 7 shows the flood extent for each different scenario on a per node basis. In addition to fast engineering application or flood risk assessment, (Bates, 2022), Bernard et al. (2022) noted that using flow metrics calculated from different precipitation rates could be used to determine the extent of flood plains and of the different channels of a river system. While more computationally demanding than geometrical method (e.g. Clubb et al., 2022), GraphFlood offers a physics-based method self-emerging the floodplain geometry. Low flow conditions in purple in Figure 7 emphasise the geometry of channel beds while higher, storm-related flow conditions in blue indicate the maximum extent of the floodplain. We only computed uniform precipitation rate scenarios, but GraphFlood can ingest spatially variable matrices of effective precipitations if coupled with more sophisticated precipitation/infiltration data or model.





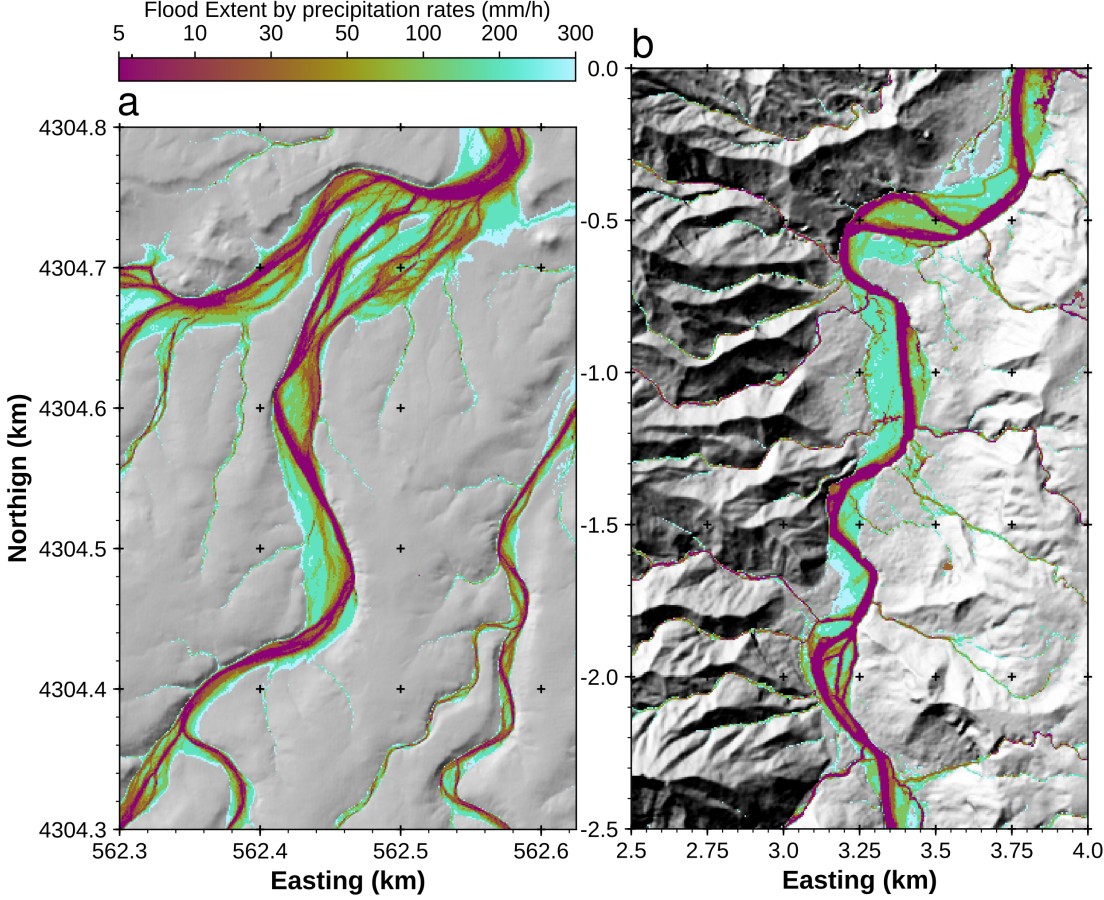

**Figure 7.** Flood extent at stationary solutions for different precipitation rates. The color represent the minimum precipitation rate at which the area is flooded by at least 10 cm of water. Note the self-emergence of bedforms and floodplains.

## 5.2 Flood wave

While the model is primarily designed and optimised for the stationary state, we illustrate its capabilities to model the transient propagation of a flood wave (e.g. sudden increase of input discharge in reach mode) in Figure 8. We isolated a small section of a river from the Green River site and started from equilibrated low flow conditions (time=0s). We instantly increase the input discharge by a factor 3 and the different panels display the spatial propagation of the resulting flood wave through time.

## 5.3 Hydromorphometry

One of the main technical challenge in topographic analysis studies is to determine from topographic data the transitions between the fluvial network, the colluvial channels, and the hillslopes. Such classification is useful for understanding landscape



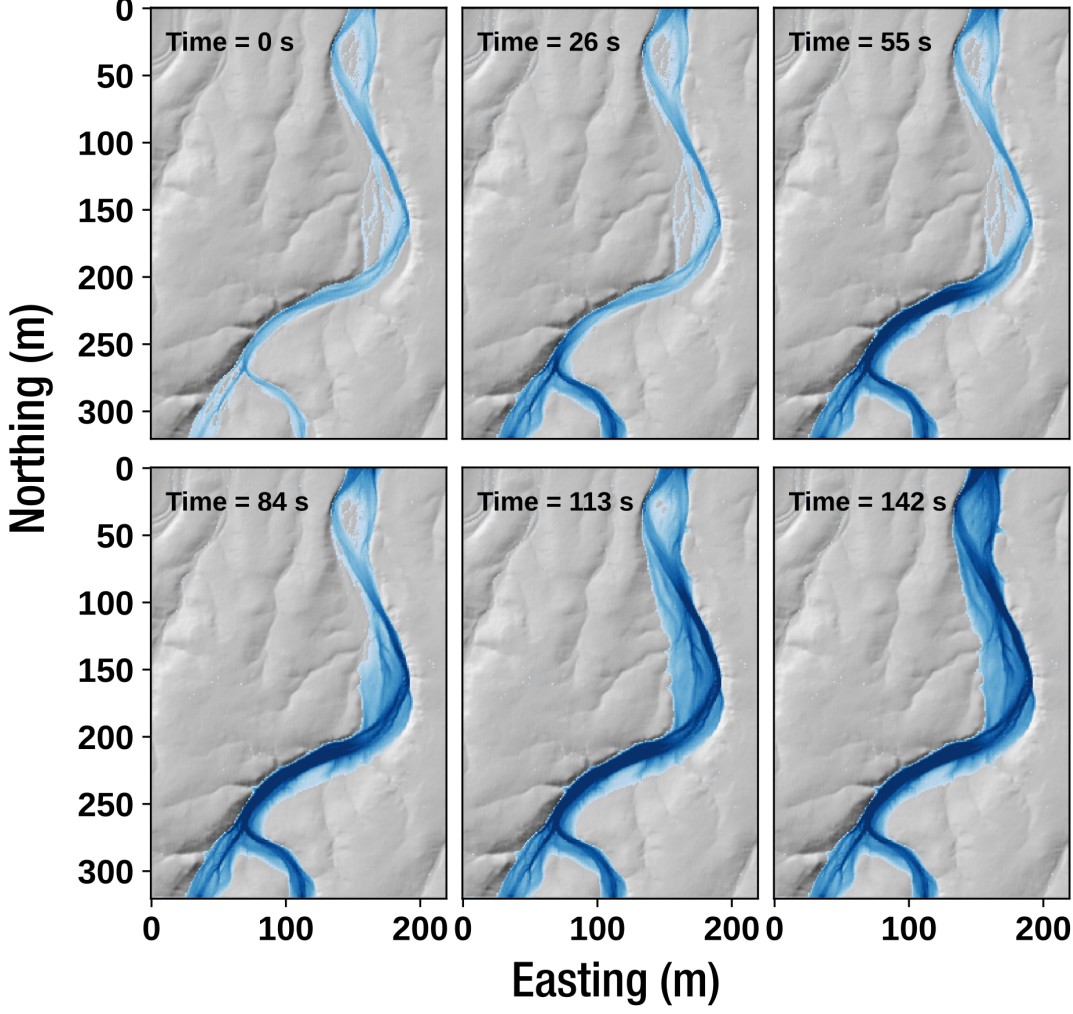

**Figure 8.** Flood extent at stationary solutions for different precipitation rates. The color represent the minimum precipitation rate at which the area is flooded by at least 10 cm of water. Note the self-emergence of bedforms and floodplains.

dynamics (e.g. Grieve et al., 2016; Hurst et al., 2019), to constrain geomorphological laws (Perron, 2011, e.g.). Landscape Evolution Models also routinely apply different process laws based on that transition (e.g. Perron, 2011), or to assess the response of landscape to tectonics or climate changes (e.g. Willett, 1999). A common approach consists in isolating breaks in the Slope-Area distributions to determine a critical drainage area value (DiBiase et al., 2010; Whipple et al., 2013, e.g.). A number of geometrical/empirical method have also been developed to isolate individual channel heads in higher resolution




DEMs (Pelletier, 2013; Clubb et al., 2014; Lurin et al., 2023, e.g.). These methods intrinsic limitation is the use of surface topography: the latter by nature cannot express the actual geometry of water bodies there making them harder to detect.

Recent studies (Costabile et al., 2019; Costabile and Costanzo, 2021; Bernard et al., 2022) demonstrated that approaches explicitly approximating hydrodynamics effectively overcome that limitation by computing hydrology-derived geomorphological metrics from hydraulic surface and discharge. They show that the slope-area relationship can incorporate hydrological information by replacing topographic slope by the hydraulic slope at equilibrium and D8 drainage area by a specific drainage area $a_s(r) = \frac{q}{r}$, where $r$ is the runoff precipitation rate and $q$ the discharge per unit width. These methods show that $a_s(r)$ is

very efficient to naturally separate river channels from colluvial channels and hillslopes. These metrics are naturally embedded within the DAG structure of GraphFlood allowing a more systematic and straightforward bulk computation. We extracted $s$ - $a_s(r)$ for both test sites and separated hillslopes, colluvial and fluvial domains (see Figure 9). For clarity, we use arbitrary thresholds from the $s$ - $a_s(r)$ plots to determine the transitions. We also define the floodplains using the maximum extent of fluvial channels for high precipitation rates from figure 7.

The $s$ - $a_s(r)$ relationships for both catchments globally show patterns similar to classic Slope-Area techniques. $s$ increases and plateaus in the hillslopes domain to then decrease with break in slopes in log space corresponding to colluvial and fluvial channels (e.g. Montgomery, 2001). However, we also observe low $s$ - $a_s(r)$ areas, corresponding to flat surfaces isolated from the channel (e.g. elevated terraces). Both sites then show a noticeable break in slope corresponding to the colluvial domain where flow starts to converge towards proto-channels, followed by another less-pronounced break in slope expressing the

switch to well define rivers domain. The addition of hydraulic information to slope and area makes the distinction less sensitive to the threshold and direct visualisation of $a_s$ give an already clear and physics-based separation of the different domains. The fluvial domains also terminate with an interesting high surge of $s$ for high $a_s(r)$ corresponding to local accelerated flow that would not be caught by common S-A plots.

      This last observation highlights the kind of additional information the hydrology-aware approach unravels. Bernard et al.

(2022) built on earlier work restricted on hillslope (Gallant and Hutchinson, 2011) where $s \equiv \frac{dz}{dx}$ to develop this principle further and express a proxy for channel width, called specific width $w_s(r)$. The specific width is calculated from the ratio between SFD drainage area (i.e. most convergent flow lines) and the specific drainage area (i.e. representing the flow field spread to its natural extent). As acknowledge by the authors, the challenge lies in the choice of the single flow path which will determine $A$: if the latter does not coincide with that main discharge field, the results are highly noisy and difficult to interpret.

With the precipiton method, Bernard et al. (2022) suggest the calculation should be post-processed on the discharge field calculated at low-flow condition and following its maximum values. We leverage GraphFlood integrated DAG data structure to optimise this process and generalise it to the 2D channel network. Indeed, using the DAG calculated from the equillibrated hydraulic surface, we repeat a stochastic walk to calculate $A$ where the steepest receivers of each nodes is determined from the multiple flow receivers using the hydraulic surface and a probability function of these receivers' $Q_{out}$. Repeating this walk

about 50 times and keeping track of node-wise $max(w_s)$ ensures all the channel pixels are visited. Figure 10 displays the resulting field of flow width where we simply apply a threshold to filter out unreasonable values happening when $A$ gets out





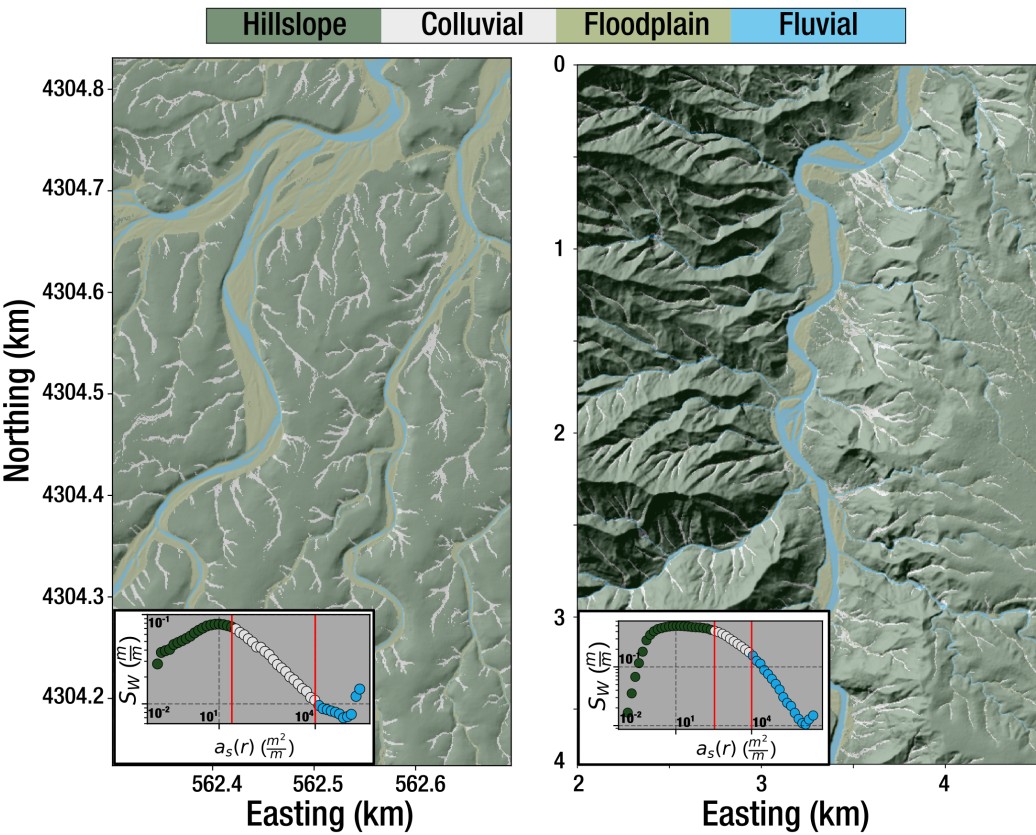

**Figure 9.** Domainification of the landscape based on hydromorphometry. Using an approch based on Bernard (2022) as well as data in figure 7, we separate the domains into area affected by hillslopes, colluvial and fluvial domains. The domains are selected by applying cutoff values on the $s - a_s(r)$ plots - see main text for details about these values. Areas that are not fluvial but flooded at high flow are considered floodplains.

of the main channel for few nodes. This method effectively highlights fine-grained variations in flow width and allows its systematic, efficient extraction unravelling patterns of "width" knickpoints.

# 6 Discussion

## 6.1 Controls on numerical efficiency and accuracy

Computational efficiency to reach the stationary solution is one of the main advantage of GraphFlood and figure 12 provides a number or benchmarks function of the number of nodes of the DEM. However, computational efficiency depends on multiple factors making the efficiency partly case dependent.



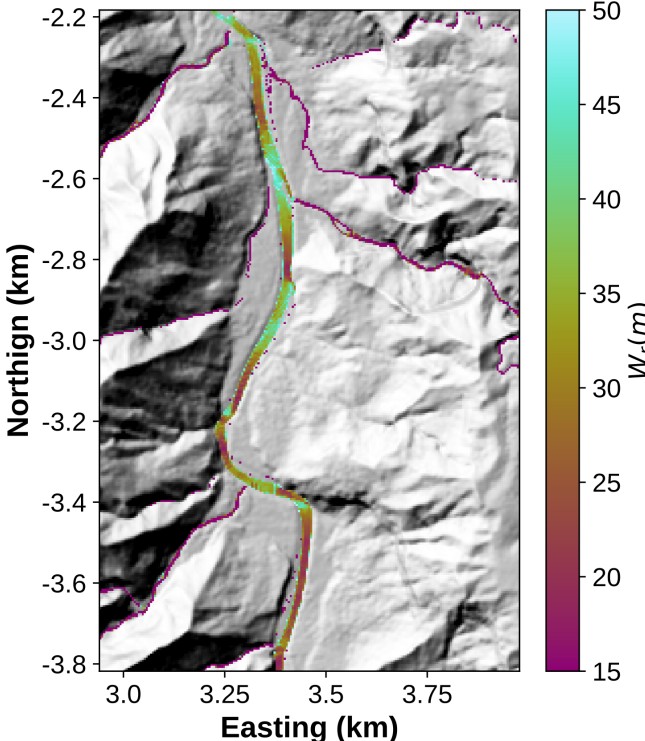

**Figure 10.** Effective width for a section of Hanalei river, reflecting channel widening and narrowing.

First, part of the method relies on subjective choices. As demonstrated on figure 5, there are spatial discrepancies in Graph-
Flood convergence speed. A study focusing on fluvial domains (e.g. flood extent) often only require <100 iterations, while
obtaining convergence for the entire landscape (e.g. separate the different process-based domains) can take up to few thou-
sands iterations. The time step also dictates the speed and accuracy of the algorithm. Maximising the time step reduces the
number of iterations to reach convergence. Yet, it also impacts the accuracy, consistency, computational time and stability of
the solution (i.e., a higher time step plateau to a fluctuating hydraulic surface).
Secondly, switching the model from MFD to SFD mode reduces the number of operations to compute and therefore the
computational time. However the resulting water surface is impacted by this choice due to the over-focusing of flow in the single
flow routing (figure 11). The line concentrating all the flow overestimates $Q_{in}$ while all the other channel nodes overestimate
$Q_{out}$ resulting in a global underestimation of $h$. The error on Green River is concentrated around 10%.
Finally the performances of GraphFlood are tightly linked to the numerical framework used for its implementation. The
simplicity and versatility of GraphFlood make it straightforward to re-implement in different frameworks as long as they of-
fer basic graph data structure and local minima handling. Computing the DAG and the related algorithms for each iteration
accounts for a big part of of the computational time. Therefore, the implementations of these algorithms strongly impact the





overall performances. For example, the exact same simulation takes approximately 250 ms or 800 ms in the python/c++ imple-
mentation or using MATLAB©/ TopoToolBox (Schwanghart and Scherler, 2014) respectively. The time consuming algorithms
are the topological ordering (e.g. Anand et al., 2020; Braun and Willett, 2013; Carretier et al., 2016), the local minima resolver
(e.g. Cordonnier et al., 2018; Barnes et al., 2014; Gailleton et al., 2023) and the receivers and donors computations as they
need updtates at each iterations.

Detailed time-benchmark comparison with other methods can also quickly be misleading because of the divergence of
scopes: GraphFlood focuses on steady flow which is conceptually too different to compare to transient solvers (e.g. Bates
et al., 2010; Brunner, 2002). River.Lab (Davy et al., 2017, formerly Floodos, ) also targets stationary solution. Bernard (2022)
demonstrated that the method could reach the same orders of magnitude for the time required to get a convergent solution in
the main rivers in specific cases where the influx of precipitons is optimised to enter only the main channel via discrete inlets
from tributary junctions. However, the efficiency of this method decreases when simulating other parts of the landscape, such
as hillslopes, due to the low frequency of precipitation passage on non-convergent areas.

Nevertheless it is worth noting the algorithm is scalable: Green River site converges in about 20 seconds for the main rivers,
with less than 200 ms per iterations. We also tested GraphFlood on an 83 Million pixels DEM on a laptop with 32 Gb of
memory and the model converged for the main rivers in about 20 hours with 100 seconds per iterations.

## 6.2  Potential optimisations

An obvious optimization consists in developing a parallel version of GraphFlood. In this paper, we made the choice to remain
on single threaded CPU for (i) simplicity, (ii) flexibility and (iii) favouring the possibility to run concurrent models to explore
parameter space. Transient mode can be parallelised, even on GPU, as each node is independent from one another at a time t
similar to Apel et al. (2022). Stationary GraphFlood, on the other hand, has a strong non-local component in the calculation of
$Q_{in}$ and would require significant modification to be partially parallelised, using for example Barnes et al. (2021) .

Another optimisation consists in improving our management of time stepping. CFL conditions only theoretically apply
to our calculation of $Q_{out}$, but not on the propagation of $Q_{in}$ in stationary mode. Alternative finite difference formulation
like Runge-Kutta or an implicit formulation could allow larger time steps. However these methods would only increase the
efficiency of a single iteration but would still suffer from the highly-iterative nature of the algorithm to reach an equilibrated
hydraulic surface.

Finally, we can significantly reduce the computation time of studies interested in the fluvial domain only. As suggested in
Bernard (2022) and illustrated in figure 4, GraphFlood converges significantly faster in areas with higher $Q$. The fluvial domain
only represents a minor subset of the total number of nodes in a landscape and theoretically, focusing only on these nodes could
significantly speed up the process. Induced sub-graph methods offer solutions to apply algorithms in a subset of a DAG without
the need to process its entirety. In the case of rivers, it requires the identification of all the nodes of interest, *i.e.* downstream of a
given discharge or drainage area threshold. Taking full advantage of this optimisation is challenging as it requires the dynamic
identifications of the nodes of interest without processing the whole graph.



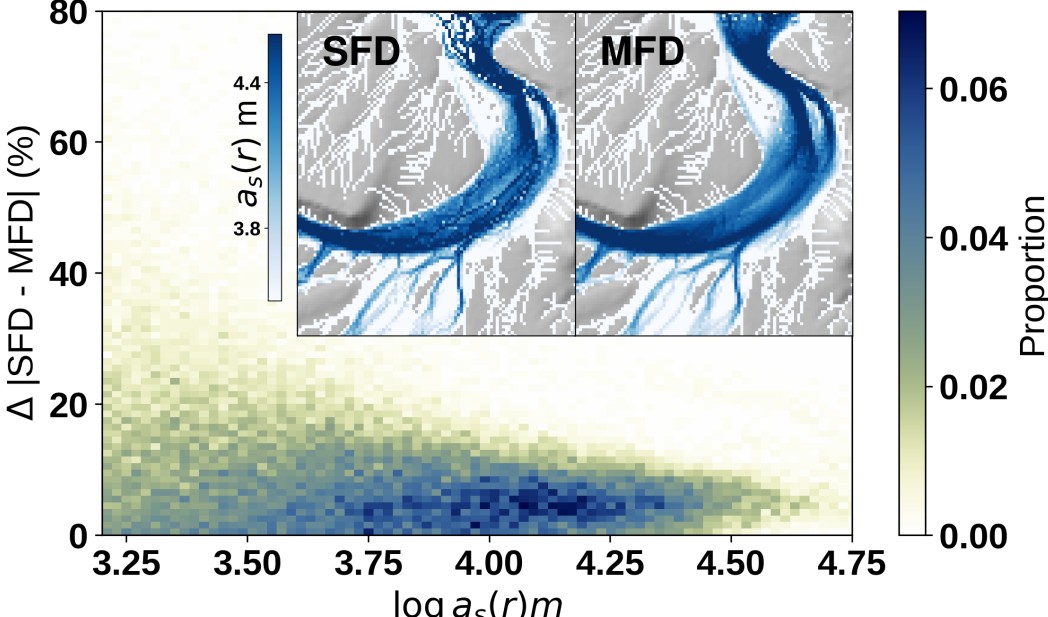

**Figure 11.** Differences in final results for Single flow solver and Multiple flow solvers. The MFD solution is cleaner and has less artifacts. The magnitude of the differences is function of the frequency at which the D8 SFD flow passes through a cell (proxied here by MFD $a_s(r)$). While SFD solvers are faster and simpler, their accuracy will be function of diverging flow patterns. Smaller $\Delta t$ can reduce the differences.

We developed an induced sub-graph method to take advantage of that optimisation. The principle remains the same than section 3.1, except that graph-realted operations are computed in a node-to-node basis (e.g. computing the DAG donors and receivers, handling of local minimas, topological ordering). A pre-computing step determines input points based on drainage area thresholds or arbitrary input points (Tarboton, 1997). These points are pushed in a priority queue sorting active nodes per decreasing elevation (opposite to Barnes et al. (2014)), ensuring that the most upstream node of interest that has not been processed yet is always the next in queue. The nodes are popped and processed from the priority queue sequentially. Once $Q_in$ and $Q_out$ computed according to section 3.1, we push in the priority queue the receivers of the active node. The process is repeate until emptying the queue. Note that if a node has no receiver and is not a model edge, we gradually fill the local depression until finding an outlet, in a similar way to Davy et al. (2017) or Gailleton et al. (2023).

This version of the algorithm reproduces the results from the original one, except minor artifacts near the input points. One iteration takes $250\,ms$ with GraphFlood and $15\,ms$ with the induced graph method. For a discharge threshold of $36000\,m^2$ and a precipitation rate of $50\,mm\,yrs^{-1}$, the models converge for the main rivers in about $50\,s$ for GraphFlood *vs* $3\,s$ for the induced graph method demonstrating strong potential for studies focusing on the fluvial domain. The complexity of the algorithm is tied to the priority queue and is therefore $\mathcal{O}(n\log n)$ with $n$ being the number of nodes in each traversal, meaning computation time



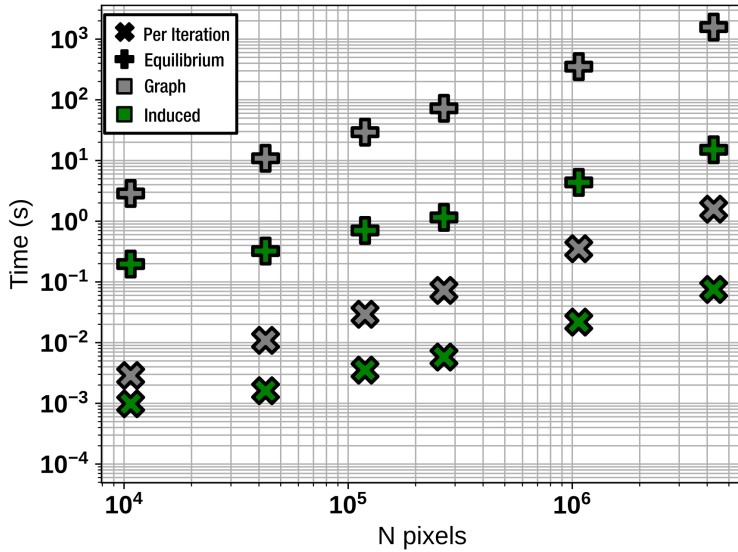

**Figure 12.** Time benchmark comparing the computational efficiency of GraphFlood and its induced graph variant for the Green River DEM resampled at various resolutions. The global convergence time represents the timing for converging the model for the fluvial and colluvial domains while the time per iterations is an important metric when considering GraphFlood for LEMs.

increases non linearly as the drainage area threshold decreases. Figure 12 provides an extensive time benchmark comparing the efficiency of both methods in a global and per-iteration perspective.

### 6.3 Potential for hydromorphometry and Landscape Evolution Models

Bernard et al. (2022) demonstrated the potential of informing common scaling laws used in tectonic geomorphology (e.g. Kirby and Whipple, 2012) with hydrodynamics. GraphFlood represents a step toward making the inclusion of hydrology more
systematic in geomorphological analysis. For example $s - a_s(r)$ plots, as illustrated by both Bernard et al. (2022) and figure 9, isolate more signals than classic $S - A$ as per originally designed by Morisawa (1962) and Flint (1974). $a_s(r)$ is not strictly function of the downstream distance like $A$ and has the potential to express a wider range of landform. Data points with high $a_s(r)$ and high $s$ are likely to represent areas of increased stream power beyond the common geometrical knickpoint (e.g., increased discharge due to local channel narrowing). Alternatively, low $s$ and $a_s$ testify of abnormally flat areas (i.e., flat areas
not visited by rivers), which if calculated from multiple runoff rates could unravel families of terraces. Commonly used metrics linked to $S - A$ (e.g., concavity index, steepness index) are likely to express a wider range of signals when extracted from $s - a_s(r)$. Combined with effective width or the direct calculation of shear stress from $h$, hydromorphometrics can help identify and quantify new family of responses to perturbations. Alongside with geometrical knickpoints (e.g. Gailleton et al., 2019), area of channel narrowing or widening or accelerated flow can be caught unravelling wider ranges of landscapes responses

to perturbations. Systematic calculations of all these metrics for multiple ranges of runoff rates could help redefining and completing global scaling laws comparing discharge, drainage area, channel width and hydraulic slopes. GraphFlood allows the fast approximation of hydrodynamics, and therefore shear stress. Coupling GraphFlood with physics based morphodynamics (e.g. Davy and Lague, 2009; Minor et al., 2022) would allow the upscaling of short term fluvial dynamics to longer time scale and larger spatial scales.

## 7 Conclusion

This study introduces GraphFlood, an efficient algorithm for solving 2D hydrodynamics based on 2D shallow water equations and specifically tailored for large DEMs. By employing Manning's equation within a graph theory framework, GraphFlood iteratively computes a stationary flow depth and discharge equilibrated to prescribed runoff rates. Leveraging graph theory algorithms ensures numerical efficiency, enabling GraphFlood to compute solutions for rivers in just seconds for a million-435 pixel DEM. Validation against analytical solutions and established models demonstrates the accuracy of GraphFlood. The simplicity, efficiency, and versatility of GraphFlood position it as a promising engine for incorporating 2D hydrodynamics into large-scale topographic analysis and landscape evolution models. Future work could utilize GraphFlood to investigate river inundation patterns, systematically extract river width as a function of water discharge, or focus on classifying landscapes to better relate landscape shape to geomorphological processes.

*Code availability.* The static version of the code used in this contribution can be found in Gailleton (2024). Updates on newer versions and more material will be posted on https://github.com/bgailleton/Gailleton_et_al_2024_GraphFlood_esurf .

*Data availability.* The DEM utilised in this study are openly available from opentopography.org under the datasets OpenTopography (2012) and OpenTopography (2020).

*Author contributions.* PS and BG designed the concept of the GraphFlood algorithms. BG, PS, PD, TB and WS designed the study. BG 445   wrote the code and ran the analysis. BG wrote the manuscript with the inputs of PS, WS, PD and TB.

*Competing interests.* Wolfgang Schwanghart is a member of the editorial board of Earth Surface Dynamics.

*Acknowledgements.* This research has been supported by the H2020 European Research Council (grant no. 803721). We thank Dimitri Lague, Guillaume Cordonnier, Ron Nativ, Fiona Clubb and Laure Guerit for constructive discussions, feedbacks and testing on GraphFlood.



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
