# Peer review of "GraphFlood 1.0: An efficient Algorithm to approximate 2D hydrodynamics for Landscape Evolution Models"

_EGUsphere, 2024_

## Referee Comment (RC2)

Review of "GraphFlood 1.0: an efficient algorithm to approximate 2D hydrodynamics for Landscape Evolution Models"

The manuscript presented a numerical hydrodynamic model which incorporates multiple flow direction of water flow on flow depth calculation with an iteration method. Computationally, the model performs better. Geologically, the model is able to produce similar morphological metrics patterns as a similar hydrodynamic model which used a single flow direction algorithm. The transient solutions of the model showing the propagation of the a flood along the channel is also interesting and has potential to be used for more applied problems.

However, I find some problems with the current manuscript that need to be improved to to clarify the model numerical and physical aspects and to sharpen the characteristics of the model.

General suggestions:

▪ One concern is about the presentation of section 2 and section 3. It leads a lot of confusion regarding the major physical and numerical structure of the GraphFlood model. After reading the Davy (2017) paper, I think this manuscript should be tailored to emphasize its difference with the Davy (2017) model while keeping the core and shared algorithms clear. If I understand correctly, the GraphFlood model is built on the skeleton of the Floodos model but imbedded the graph topological ordering of nodes for water depth calculation with iteration when the water flow direction algorithm is multiple or single flow direction. The iteration method is clearly presented with the chart between line 188 and line 189 and Figure 2 (although only for the D8 single flow direction). However, the multiple flow direction algorithm in the context of this model is not presented at all, or only minorly with Figure 1 but with very limited captions or explanations in the text to understand. I think a comparable figure, similar to Figure 2, showing the multiple flow direction should be made to sharpen the point of the GrahFlood model.

Section 2 and section 3 used many symbols in equations, text, and figure captions. However, they should be cleaned up to keep consistency and should be explained either in the text or in a summarization table. Right now, different symbols representing same physical parameters are used in the text and equations. For example, the water depth is represented by $h$ in the beginning but later $hw$ is used in figures, text, and equations. Some symbols are missing explanation and they appear in many places in the manuscript. I identified some and pointed them out in the line-by-line comments below. Note that section 4 also has the similar problems of symbols inconsistency. Please check it through as well.

▪ The other more scientific question is about application to hydromorphometry which is presented in Section 5.3 and discussed further in section 6.3. I find it hard to relate what's

being described in the text of with the supporting figure, Figure 9. The curve segments described in lines 320-323 needs to be indicated on Figure 9 properly. Also, segments indicating different domains were divided with arbitrary cut-off values, why so? Except for the confusion due to presentation style, one scientific question is what is different compared with the Bernard et al (2022) model where single flow direction is used? And another question is how to understand the similarity of s-a(r) curves for the two very different sites where one is flat with low relief, maybe depositional and the other site is steep hillslopes and bedrock channels? These two questions can shed light to landscape evolution modellings of geological scales and tectonic geomorphology studies as also mentioned by the authors. I think this part should be expanded further to demonstrate the important applications of this model. Maybe further analysis is needed to do so, for example, making the s-a(r), or w(r) analysis on the same study site of the Bernard paper with the GraphFlood model. Given that the GraphFlood model outpaces the other models in terms of computation efficiency, it will be better to further demonstrate its geological applicability in terms of the multiple flow direction realization of hydrodynamic laws.

Some minor problems to be fixed:

- Line 52: "…represented by one pixel-wide paths (Figure 2).", Figure 1 should be referred to here.
- Figure 1 captions and key: 1) it is not clear what the black and red arrows in c and d are showing which parameter(s) vector(s). Are they flow velocities showing the relative rate and direction? 2) the symbols, $H_w$, in a/c and the $A$ in b/d panel need explicit explanation in the figure caption.
- Line 80: "Precipitons increase the water height along their path, bypassing the need to to propagate", typo here, to to
- Line 88-line 90: "A similar approach has been … our new algorithm". I don't see how the multiple flow algorithm suddenly jumps in here after the statement of information integration along 1D flow path. Could you expand to provide more connections with the previous sentence? Also, the abbreviation of multiple flow direction (MFD) should be mentioned here before use it.
- Line 92: GraphFlood is the name of the proposed new model here. It appears in the manuscript for the first time. I suggest making a clear statement that the model is referred to as "GraphFlood" before using it here. Probably the statement should be made even before Figure 1 because Figure 1 used this term already.
- Equation (2), what is $s$ in the equation? Line 119, what is $xmax$? Seems $s$ is a vector describing the slope (topographic or hydraulic? If hydraulic, what is the exact definition?) of the direction of the steepest hydraulic gradient. Please double check the equation and the explanation of symbols here.

- Line 125: "…the indices **Xin** and **Xout** to refer…", is it better to say "…the subscript *in* and *out* to refer…"? X here can be mistaken as a new parameter.
- Line 139-141: "In the following, we detail the numerical graph…, we describe the finite …, explain … and validate…". Need grammar check.
- Line 146: "hydraulic surface ($Z + h$)", the symbol $Z$ needs to be explained. $Z$ is topographic elevation?
- Line 150: "…to downstream and *sink filling* a method filling local …",  Need grammar check.
- Line 153: "…or Multiple Flow Direction (MFD) DAGs (e.g. Tarboton, 1997; Anand et al., 2020).", there should be a short explanation of multiple flow direction in terms of node and receiver here.
- Line 162-163: "One advantage of GraphFlood is that it can be implemented using existing computational frameworks for DEM analysis and LEM simulation …", meaning the flow direction indexing with the topologically sorted order?  More specific details should be provided here.
- Equation (6): the symbol $S_{ijmax}$ need to be explained in the text. It is the maximum slope between the node $i$ and all its receivers?
- Line 185: "The overall process is outlined on algorithm 1.", should the algorithm 1 chart be indexed as tables or figures for clearer citation here? Plus, the symbol $h_w$ in the chart of algorithm 1 is not explained in the text.
- Line 200: "…depth becomes lowerthan an acceptable…", a space is missing between lower than.
- Line 194: "Equation 6 expresses the velocity of a flood wave…", should it be Equation 6 and equation 3 express the velocity of flood wave?
- Figure 2: 1) caption doesn't match with the figure at all. 2) figure 2 should be cited properly in the text, perhaps in section 3.4. 3) $Q_w$ means water flux? The symbol is not explained in the text. $hw$ is water depth? If sediment flux is not talked at all in the equations or text, I suggest take out the subscript *w* from $Q$ and $h$. 4) it is not clear how iteration 2 is related with iteration 1.1 and 1.2 from the figure. Caption is needed to explain it.
- Line 219 and equation (9): use h* or hw*?
- Figure 3: Will it be better to point out the stationary and transient (numerical) solution curves on figure a? Or explain them more clearly in the captions. Or use line styles or colors to distinguish them.
- Line 274: typo "previoous"
- Line 288: "…flood risk assessment, (Bates, 2022), Bernard et al. (2022) noted that…", both papers made the point?
- Figure 8: need to include the colorbar in the figure. Dark blue is bigger water depth and light blue is lower water depth? It is confusing without a clear colorbar indicating the parameters being colored on the maps.

- Line 298-299: "We instantly increase the input discharge by a factor 3 and …", does it mean that higher input discharge lasts for all time steps from time 1 to time end? It is a bit unclearly stated here especially compared with figure 8 which shows higher water depth everywhere but migrating flooding extent from time 26s to time 142s. I feel section 5.2 under-described these observations.
- Line 314: I feel the Bernard et al (2022) paper should be cited here for a clear reference of the specific drainage area, $a_s(r)$.
- Line 318 and Figure 9: "For clarity, we use arbitrary thresholds from the s - as(r) plots to determine the transitions.", 1) what is s? the hydraulic slope or topographic slope? 2) I don't understand why arbitrary thresholds are used in Figure 9 to divide the different domains…

---

## Author Comment (AC1)

**Response to Reviewers' Comments on**

*GraphFlood 1.0: an efficient algorithm to approximate 2D hydrodynamics for Landscape Evolution Models*

B. Gailleton P. Steer P. Davy W. Schwanghart T. Bernard

August 30, 2024

In the following text, line numbers in the reviewer comments refer to the original manuscript, while line numbers in our responses correspond to the diff file.

**Reviewer 1 Comments**

**Reviewer Comment:** *The manuscript by Gailleton et al. presents a suite of algorithms designed to solve the shallow water equation over gridded topographic data, utilizing graph theory-related data structures and associated tree scanning algorithms. By doing so, the authors leverage advancements and overcome some challenges of two existing approaches for hydrologic and landscape evolution calculations. Specifically, the authors combine (1) recent advances in applying graph theory-related algorithms in large-scale Landscape Evolution Models, which speed up computations of drainage area and consequently erosion/sedimentation rates (e.g., Braun and Willett, 2013) but commonly assume that flow occurs as 1D lines, and (2) shallow water equation solvers that more realistically represent flow routing over complex topography but commonly suffer from high computational costs, limiting their usage to small length scales and short time scales.*

**Response:** We thank Reviewer #1 for reviewing our contribution and for their positive comments and constructive suggestions. Please find bellow our responses to the different concerns.

**Reviewer Comment:** *The presented numerical advances are impressive and substantial. The manuscript makes convincing arguments that their utility could be fundamental for specific hydrologic and landscape evolution problems. However, the manuscript remains in the realm of model development and, as such, doesn't provide new insights into natural processes (including what models' emergent dynamics teach us about how nature could work). Consequently, the manuscript reads more like (an impressive) technical report. If expanding the analysis toward more physical insights is desired, the current manuscript holds some threads that could be pulled to produce new, insightful understandings.*

**Response:** We appreciate Reviewer #1's observation regarding the technical focus of our contribution and acknowledge that our work emphasizes the method rather than natural processes. However, we argue that technical papers are essential as they lay the groundwork for introducing new methods. First, they enhance the reproducibility of scientific contributions by providing detailed descriptions of their implementations, enabling different frameworks to utilize or adapt them (for example, we are currently implementing the method in TopoToolBox - [9]). Furthermore, future research focusing on natural processes will benefit from the ability to cite

the method in full detail, without relying on shortcuts, hiding limitations or relegating essential information to supplementary materials. Lastly, ESURF already includes several method-heavy papers (e.g., [8, 3, 6]) that have served as foundations for subsequent scientific applications (e.g., [5, 2, 7]).

**Reviewer Comment:** *For example, the analysis presented in figure 7 could potentially be utilized to produce preliminary (and possibly case-specific) insights about the relationship of flow width as a function of bedforms and local relief (and precipitation input, of course). There are more opportunities like this in the manuscript.*

**Response:** In response to Reviewer #2, we have also increased the amount of applied analysis in section 5. Notably, we observed that both our test sites and those studied by Bernard et al. (2022) show similarities in their global patterns, but with significantly different values and local trends in $s$ - $a_s(r)$ plots. This underscores the need, that we will address in a future study, for more systematic and larger scale analyses, to uncover consistent trends and relate them to river morphology, tectonics, climate or lithology for example. The related changes are mainly in Section 5.3 and 6.3.

**Line-by-line comments**

**L. 53-54:** *Lines 53-54 list several studies demonstrating the advantages of integrating 2D hydrodynamics to inform the study of landforms. What is missing in these studies?*

**Response:** These studies introduce the use of sophisticated hydrodynamics methods for analyzing landscapes, they do not "miss" anything in themselves but the method they use are challenging to upscale - which is the goal of GraphFlood. We adapted the text to better reflect these ideas.

**L. Fig. 1:** *What are the arrows? How are they scaled?*

**Response:** The arrows represent the flow velocity vector scaled by magnitude. We appended the figure caption. See lines 67-79.

**L. 70:** *What does "remains hampered by the physics behind which explicitly simulates wave propagation" mean?*

**Response:** This sentence refers to CAESAR-LISFLOOD shallow water approximation which simulates the time-transient propagation of flow depth and discharge. We clarified the text. See lines 67-79.

**L. 120:** *The text about xmax is out of context.*

**Response:** We removed the sentence, it was a remnant of a former notation.

**L. 146:** *Unsure if z was defined?*

**Response:** $Z$ is the topographic surface, we homogenised the notations and added a table of definitions in appendix A1.

**L. 164:** *Since the hydraulic surface is a crucial concept, readers might benefit from a formal definition and possibly a demonstration (relative to topography).*

**Response:** We added a clearer definition of the hydraulic surface. after equation 1 (lines 147-149).

**L. 182:** *"The magnitude of Qout flux is the same for MFD and SFD schemes." Is this an assumption (represented by the correction factor) or a fact? Is it also valid for the transient case? Since this is such a central corollary, perhaps the authors can expand on this issue.*

**Response:** The solution is the same for both SFD and MFD which are just different numerical approximation of the same equation. They show a small difference linked to the partitioning $Q_{in}$ (¡10% as demonstrated on figure 11 on the new manuscript). This is valid for both transient and static solutions.

**L. 207:** *What does "while in reach mode, given entry nodes receive an arbitrary Qin" mean?*

**Response:** Precipitations and reach are two different ways of informing GraphFlood's water discharge input. Reach mode means boundary nodes belonging to the upstream section of the river receive water while precipitation mode gives water to every cells. We clarified the text lines 245-246.

**L. Fig 2.:** *Can't understand the comparison. Which of the panels are a and c?*

**Response:** Following comments from reviewer #1 and #2, we reworked Figure 2 entirely.

**L. 215:** *What does "(then determined in respect to CFL conditions)" mean?*

**Response:** This sentence was confusing and did not help to the comprehension of the method. We removed it.

**L. Fig. 3.:** *It is possible to guess from the context, but readers will benefit from using different symbols for the stationary and transient solutions. I would also be interested in the actual time represented by these simulations (perhaps multiple time axes are needed). It is interesting to understand the time necessary for the transient solution to converge to the analytic solution, as this represents (gives a hunch for) the time scale over which the transient solution is important. Additionally, consider adding a graphical representation of the rectangular channel.*

**Response:** We modified the figure to clarify the transient *vs* static solutions. We also incorporated information about cpu timing in the caption, as well as physical timing required for the transient model to converge to the final solution.

**L. Fig. 8.:** *Wrong caption. Duplication of fig 7.*

**Response:** We corrected the caption.

**L. Fig. 9:** *S as. The interpretation of the colluvial sections is surprising. Are the white pixels (channels) in the left-hand side, lower relief landscape truly colluvial? The different types of slope break from colluvial to fluvial (lower slope on the LHS and higher slope on the RHS) could indicate that the interpretation should be more complex. Additionally, the slope increase*

*at the high S sa end is surprising. While mentioned in line 327, its source is still not clear.*

**Response:** In response to this and Reviewer #2's main comment, we significantly reworked Figure 9 and the associated text to better explain and emphasize our classifications. To avoid confusion with the sedimentological connotations of the term, we no longer refer to the "colluvial" domain. Additionally, we included a new figure in the appendix that isolates and illustrates the high slope/high specific drainage area points, accompanied by detailed text explaining their significance.

**L. 412:** $Q_{out}$ and $Q_{in}$

**Response:** We corrected the typo.

**L. all:** *Consider reducing the use of acronyms. Acronyms are useful for the authors but less for casual readers (specifically those not from the same field).*

**Response:** We significantly reduced the number of acronyms. We only kept cross-disciplinary ones: DEM (digital elevation model), LEM (Landscape Evolution Model) or CFL (Courant-Friedrich Levy).

**Reviewer 2 comments:**

**Reviewer Comment:** *The manuscript presented a numerical hydrodynamic model which incorporates multiple flow direction of water flow on flow depth calculation with an iteration method. Computationally, the model performs better. Geologically, the model is able to produce similar morphological metrics patterns as a similar hydrodynamic model which used a single flow direction algorithm. The transient solutions of the model showing the propagation of the a flood along the channel is also interesting and has potential to be used for more applied problems.*

**Response:** We thank reviewer #2 for his constructive and thorough review of our work that have clearly helped improving the quality of our manuscript. Bellow, our responses and subsequent modifications to their comments and concerns.

**Reviewer Comment:** *However, I find some problems with the current manuscript that need to be improved to to clarify the model numerical and physical aspects and to sharpen the characteristics of the model. One concern is about the presentation of section 2 and section 3. It leads a lot of confusion regarding the major physical and numerical structure of the GraphFlood model. After reading the Davy (2017) paper, I think this manuscript should be tailored to emphasize its difference with the Davy (2017) model while keeping the core and shared algorithms clear. If I understand correctly, the GraphFlood model is built on the skeleton of the Floodos model but imbedded the graph topological ordering of nodes for water depth calculation with iteration when the water flow direction algorithm is multiple or single flow direction. The iteration method is clearly presented with the chart between line 188 and line 189 and Figure 2 (although only for the D8 single flow direction). However, the multiple flow direction algorithm in the context of this model is not presented at all, or only minorly with Figure 1 but with very limited captions or explanations in the text to understand. I think a comparable figure, similar to Figure 2, showing the multiple flow direction should be made to sharpen the point of the GrahFlood model.*

**Response:** We appreciate Reviewer #2's request for clarification on our methods and on the

similarities and differences with Floodos (Davy et al., 2017). In Section 2, we present the theoretical and physical background underlying our approximation of the shallow water equations. This section closely mirrors Floodos because both methods aim to solve the same governing equations. While both methods can model the transient propagation of flood waves, they are particularly effective under the steady-flow assumption, which allows for a more efficient determination of the stationary solution. However, the numerical schemes, algorithms, scalability, ease of re-implementation, and overall efficiency differ significantly between the two.

The primary philosophical difference lies in the global approach of our solver. At the start of each iteration, GraphFlood calculates the global connectivity of the entire terrain using a directed acyclic graph, employing either single or multiple flow directions (i.e., transmitting the water flux to a single steepest descent receiver or partitioning it among multiple downstream nodes). This graph data structure allows GraphFlood to calculate $Q_{in}$ and $Q_{out}$ for the entire landscape before incrementing $h$ everywhere simultaneously and proceeding to the next iteration. The single flow is straightforward to implement and more efficient in term of computation ressources than the multiple flow algorithm, but less accurate as demonstrated on the new figure in the appendix.

In contrast, Floodos operates differently by sequentially dropping particles (precipitons) onto the landscape. These precipitons follow a stochastic path on the hydraulic surface, each with a unique timestamp. As they traverse the landscape, they directly affect the flow depth of the cell they encounter before moving to the next location. Floodos does not explicitly calculate $Q_{in}$ like GraphFlood. Instead, precipitons deposit a constant volume determined by the total field of water influx and balanced by manning's friction equation. Each precipiton has its own timestamp resulting in local optimised and spatially variable time steps. While a single precipiton follows a 1D path, the cumulative effect of multiple stochastic paths simulates something akin to multiple flow directions, though it cannot be strictly described in terms of graph theory (i.e. single vs multiple flow direction).

Floodos is particularly effective in fluvial valleys or areas with convergent flow routing due to the high frequency of precipiton passage, making it well-suited for estimating shallow water equations in such domains. However, this frequency decreases significantly in areas with lower drainage, leading to slower convergence (see Chapter 5 of [1] for an example). GraphFlood's global approach addresses this issue by systematically processing every node, maintaining efficiency through the use of selected graph theory algorithms.

To clarify the manuscript and address this comment:

- We reworked the text introducing Floodos in Section 1 ("existing methods") to better reflect the important principles, pros and cons behind their method. We removed unnecessary technical details. See lines 67 to 105.

- We reworked entire paragraphs of the subsection "A new solution based on graph theory" in order to clarify the reasons behind GraphFlood's development and its differences from existing methods - in particular from Floodos. See lines 107-132. We also modified the first paragraphs of section 2 and 3 in the same direction. See lines 135-140 and 165-170.

- We modified Figure 2, which now explicitly illustrates both single and multiple flow versions of the algorithm.

**Reviewer Comment:** *Section 2 and section 3 used many symbols in equations, text, and figure captions. However, they should be cleaned up to keep consistency and should be explained either in the text or in a summarization table. Right now, different symbols representing same physical parameters are used in the text and equations. For example, the water depth is represented by h in the beginning but later hw is used in figures, text, and equations. Some symbols*

*are missing explanation and they appear in many places in the manuscript. I identified some and pointed them out in the line-by-line comments below. Note that section 4 also has the similar problems of symbols inconsistency. Please check it through as well.*

**Response:** We agree with this comment and have, in turn, homogenised the notations and thoroughly checked their consistency in the text/figures. Also in response to reviewer #1, we added a appendix with a table for the different notations.

**Reviewer Comment:** *The other more scientific question is about application to hydromorphometry which is presented in Section 5.3 and discussed further in section 6.3. I find it hard to relate what's being described in the text of with the supporting figure, Figure 9. The curve segments described in lines 320-323 needs to be indicated on Figure 9 properly. Also, segments indicating different domains were divided with arbitrary cut-off values, why so? Except for the confusion due to presentation style, one scientific question is what is different compared with the Bernard et al (2022) model where single flow direction is used? And another question is how to understand the similarity of s-a(r) curves for the two very different sites where one is flat with low relief, maybe depositional and the other site is steep hillslopes and bedrock channels? These two questions can shed light to landscape evolution modellings of geological scales and tectonic geomorphology studies as also mentioned by the authors. I think this part should be expanded further to demonstrate the important applications of this model. Maybe further analysis is needed to do so, for example, making the s-a(r), or w(r) analysis on the same study site of the Bernard paper with the GraphFlood model. Given that the GraphFlood model outpaces the other models in terms of computation efficiency, it will be better to further demonstrate its geological applicability in terms of the multiple flow direction realization of hydrodynamic laws.*

**Response:** Regarding the differences with Bernard et al. (2022), we believe we have already demonstrated in our response to previous comments that both methods yield comparable results. Since we already compare both algorithms on our Green River sites in another section, we believe reapplying the same analysis to the same site as in Bernard et al. (2022) would not significantly enhance our contribution. However, we have made substantial revisions to Section 5.3 to clarify and expand upon the $s$ - $a_s(r)$ data analysis.

First, we refined the classification of the different domains to align more closely with Bernard et al. (2022). We now differentiate only three domains— I, II, and III—consistent with Figure 6 in their work. We have also refined our thresholds, now corresponding to clear breaks in slope on the $s$ - $a_s(r)$ data. We removed the floodplain class, which was based on flood extents, obtained for different rainfall rate, already depicted in Figure 7. The different domains are now consistently represented in Figure 9 a and b in map view, along with the linear regressions that demonstrate the breaks in slope on the $s$ - $a_s(r)$ in panel c and d.

Additionally, we introduced a new class of "outliers" corresponding to the data points in the final section of the $s$ - $a_s(r)$ plots, which are not present in classic Slope-Area methods. These outliers have been isolated and plotted in a new figure (Figure 10) to illustrate their geomorphological significance.

We have substantially revised the associated text in Section 5.3. Reviewer #2 also raised the issue of similarities and differences between the different sites. While both sites exhibit the general domains I, II, and III common to Slope-Area techniques, they also show very different values, coefficients, intercepts, and local trends. We agree that this is a promising research avenue, but we feel that relating these differences to process would require a lot more analysis and we do not wish to over-interpret from 2 to 3 test sites. We suggest that the variations of hydraulic-informed $s$ - $a_s(r)$ metrics would benefit further from dedicated studies with a broader scope to identify consistent trends in large-scale landscapes with known climatic, tectonics or

lithologic constrains (some of which are already in preparation). We have added a sentence in the discussion to emphasise this direction. See lines 488 - 500.

**Line-by-line comments:**

**L. Line 52::** *". . . represented by one pixel-wide paths (Figure 2).", Figure 1 should be referred to here.*

**Response:** Done.

**L. Figure 1 captions:** *and key: 1) it is not clear what the black and red arrows in c and d are showing which parameter(s) vector(s). Are they flow velocities showing the relative rate and direction? 2) the symbols, Hw, in a/c and the A in b/d panel need explicit explanation in the figure caption.*

**Response:** We clarified the caption of figure 1.

**L. Line 80::** *"Precipitons increase the water height along their path, bypassing the need to to propagate", typo here, to to*

**Response:** We rephrased that section. See lines 80 - 100.

**L. Line 88-line:** *90: "A similar approach has been . . . our new algorithm". I don't see how the multiple flow algorithm suddenly jumps in here after the statement of information integration along 1D flow path. Could you expand to provide more connections with the previous sentence? Also, the abbreviation of multiple flow direction (MFD) should be mentioned here before use it.*

**Response:** We removed the use of the abbreviation MFD in response to reviewer #1 (niche abbreviations reduce readability). We rephrased and restructured the paragraph to clarify. See lines 80 - 100.

**L. Line 92::** *GraphFlood is the name of the proposed new model here. It appears in the manuscript for the first time. I suggest making a clear statement that the model is referred to as "GraphFlood" before using it here. Probably the statement should be made even before Figure 1 because Figure 1 used this term already.*

**Response:** We modified the end of the very first section to explicitly introduce GraphFlood. See line 57.

**L. Equation (2):** *What is s in the equation? Line 119, what is xmax? Seems s is a vector describing the slope (topographic or hydraulic? If hydraulic, what is the exact definition?) of the direction of the steepest hydraulic gradient. Please double check the equation and the explanation of symbols here.*

**Response:** We clarified the consistence of the notations as well as the text around equation 2. xmax was a remnant of a former notation. See line 147-150.

**L. Line 125::** *". . . the indices Xin and Xout to refer. . . ", is it better to say ". . . the subscript in and out to refer. . . "? X here can be mistaken as a new parameter.*

**Response:** We edited the sentence. See line 149.

**L. Line 139-141:** *: "In the following, we detail the numerical graph..., we describe the finite ..., explain ... and validate...". Need grammar check.*

**Response:** Done.

**L. Line 146::** *"hydraulic surface (Z + h)", the symbol Z needs to be explained. Z is topographic elevation?*

**Response:** We clarified the notations and added a table of nomenclature.

**L. Line 150::** *"...to downstream and sink filling a method filling local ...", Need grammar check.*

**Response:** Done.

**L. Line 153::** *"...or Multiple Flow Direction (MFD) DAGs (e.g. Tarboton, 1997; Anand et al., 2020).", there should be a short explanation of multiple flow direction in terms of node and receiver here.*

**Response:** We added a brief definition of multiple flow direction in the previous paragraph (first mention of multiple flow direction). See line 185.

**L. Line 162-163:** *: "One advantage of GraphFlood is that it can be implemented using existing computational frameworks for DEM analysis and LEM simulation ...", meaning the flow direction indexing with the topologically sorted order? More specific details should be provided here.*

**Response:** That is correct. We appended the text to clarify. See line 198.

**L. Equation (6):** *: the symbol $S_{ij}max$ need to be explained in the text. It is the maximum slope between the node i and all its receivers?*

**Response:** We added a description on the previous equation (first mention). See line 216.

**L. Line 185::** *"The overall process is outlined on algorithm 1.", should the algorithm 1 chart be indexed as tables or figures for clearer citation here? Plus, the symbol $h_w$ in the chart of algorithm 1 is not explained in the text.*

**Response:** We homogenised the notations. We believe that our pseudo-algorithm is formatted in the usual way (e.g. [4]).

**L. Line 200::** *"...depth becomes lowerthan an acceptable...", a space is missing between lower than.*

**Response:** Fixed.

**L. Line 194::** *"Equation 6 expresses the velocity of a flood wave...", should it be Equation 6 and equation 3 express the velocity of flood wave?*

**Response:** We were indeed referring to equation 3 and not 6, thanks for noticing. We fixed the ref.

**L. Figure 2::** *1) caption doesn't match with the figure at all. 2) figure 2 should be cited properly in the text, perhaps in section 3.4. 3) Qw means water flux? The symbol is not explained in the text. hw is water depth? If sediment flux is not talked at all in the equations or text, I suggest take out the subscript w from Q and h. 4) it is not clear how iteration 2 is related with iteration 1.1 and 1.2 from the figure. Caption is needed to explain it.*

**Response:** See our responses to the main comments.

**L. Line 219 and:** *equation (9): use h* or hw*?*

**Response:** We reviewed the whole notation system to be more consistent.

**L. Figure 3::** *Will it be better to point out the stationary and transient (numerical) solution curves on figure a? Or explain them more clearly in the captions. Or use line styles or colors to distinguish them.*

**Response:** We added labels on the figure to clarify.

**L. Line 274::** *typo "previoous"*

**Response:** Sorted.

**L. Line 288::** *"...flood risk assessment, (Bates, 2022), Bernard et al. (2022) noted that...", both papers made the point?*

**Response:** (Bates, 2022) is a review about the use of reduced complexity model for flood risk assessment - referring to the beginning of the sentence while Bernard et al. (2022) made the observations about flow metrics. We added "e.g." to clarify the flow.

**L. Figure 8::** *need to include the colorbar in the figure. Dark blue is bigger water depth and light blue is lower water depth? It is confusing without a clear colorbar indicating the parameters being colored on the maps.*

**Response:** We added the colorbar to the figure.

**L. Line 298-299:** *: "We instantly increase the input discharge by a factor 3 and ...", does it mean that higher input discharge lasts for all time steps from time 1 to time end? It is a bit unclearly stated here especially compared with figure 8 which shows higher water depth everywhere but migrating flooding extent from time 26s to time 142s. I feel section 5.2 under-described these observations.*

**Response:** We clarified the caption of the figure to provide all the necessary details.

**L. Line 314::** *I feel the Bernard et al (2022) paper should be cited here for a clear reference of the specific drainage area, as(r).*

**Response:** Done.

**L. Line 318 and:** *Figure 9: "For clarity, we use arbitrary thresholds from the s - as(r) plots to determine the transitions.", 1) what is s? the hydraulic slope or topographic slope? 2) I don't understand why arbitrary thresholds are used in Figure 9 to divide the different domains...*

**Response:** See our response to the main comment.

**References**

[1] Thomas Bernard. *Analyse haute résolution de la morphologie des paysages et des processus à partir de LiDAR aéroporté répété et simulation hydraulique.* These de doctorat, Rennes 1, March 2022.

[2] Fiona J. Clubb, Simon M. Mudd, Taylor F. Schildgen, Peter A. van der Beek, Rahul Devrani, and Hugh D. Sinclair. Himalayan valley-floor widths controlled by tectonically driven exhumation. *Nature Geoscience*, 16(8):739–746, August 2023. Number: 8 Publisher: Nature Publishing Group.

[3] Fiona J. Clubb, Eliot F. Weir, and Simon M. Mudd. Continuous measurements of valley floor width in mountainous landscapes. *Earth Surface Dynamics*, 10(3):437–456, May 2022. Publisher: Copernicus GmbH.

[4] Guillaume Cordonnier, Benoît Bovy, and Jean Braun. A Versatile, Linear Complexity Algorithm for Flow Routing in Topographies with Depressions. *Earth Surface Dynamics Discussions*, 7(2):1–18, 2018. Publisher: Copernicus GmbH.

[5] Boris Gailleton, Simon M. Mudd, Fiona J. Clubb, Stuart W. D. Grieve, and Martin D. Hurst. Impact of Changing Concavity Indices on Channel Steepness and Divide Migration Metrics. *Journal of Geophysical Research: Earth Surface*, 126(10):e2020JF006060, 2021. _eprint: https://onlinelibrary.wiley.com/doi/pdf/10.1029/2020JF006060.

[6] Stefan Hergarten. Transport-limited fluvial erosion – simple formulation and efficient numerical treatment. *Earth Surface Dynamics*, 8(4):841–854, October 2020. Publisher: Copernicus GmbH.

[7] Stefan Hergarten. Theoretical and numerical considerations of rivers in a tectonically inactive foreland. *Earth Surface Dynamics*, 10(4):671–686, July 2022. Publisher: Copernicus GmbH.

[8] Simon M. Mudd, Fiona J. Clubb, Boris Gailleton, and Martin D. Hurst. How concave are river channels? *Earth Surface Dynamics*, 6(2):505–523, 2018.

[9] W. Schwanghart and D. Scherler. Short Communication: TopoToolbox 2 - MATLAB-based software for topographic analysis and modeling in Earth surface sciences. *Earth Surface Dynamics*, 2(1):1–7, 2014.